# Neural Sampling from Boltzmann Densities: Fisher-Rao Curves in the Wasserstein Geometry

**Jannis Chemseddine, Christian Wald, Richard Duong & Gabriele Steidl**
Institute of Mathematics
TU Berlin
Straße des 17. Juni 136 Berlin, Germany
{chemseddine,wald,duong,steidl}@math.tu-berlin.de

## Abstract

We deal with the task of sampling from an unnormalized Boltzmann density $\rho_D$ by learning a Boltzmann curve given by energies $f_t$ starting in a simple density $\rho_Z$. First, we examine conditions under which Fisher-Rao flows are absolutely continuous in the Wasserstein geometry. Second, we address specific interpolations $f_t$ and the learning of the related density/velocity pairs $(\rho_t, v_t)$. It was numerically observed that the linear interpolation, which requires only a parametrization of the velocity field $v_t$, suffers from a "teleportation-of-mass" issue. Using tools from the Wasserstein geometry, we give an analytical example, where we can precisely measure the explosion of the velocity field. Inspired by Máté and Fleuret, who parametrize both $f_t$ and $v_t$, we propose an interpolation which parametrizes only $f_t$ and fixes an appropriate $v_t$. This corresponds to the Wasserstein gradient flow of the Kullback-Leibler divergence related to Langevin dynamics. We demonstrate by numerical examples that our model provides a well-behaved flow field which successfully solves the above sampling task.

## 1 Introduction

In this paper, we consider the problem of sampling from a Boltzmann density $\rho_D = e^{-f_D}/Z_D$ with unknown normalizing constant $Z_D$. Our approach is in the spirit of recent developments in generative modelling and aims to construct a curve $\rho_t = e^{-f_t}/Z_t$ interpolating between a simple density $\rho_Z$ and the target $\rho_D$, i.e. $\rho_0 = \rho_Z$ and $\rho_1 = \rho_D$. If such curve admits a velocity field $v_t$ and there exists a solution $\varphi$ of

$$\partial_t \varphi_t = v_t(\varphi_t), \quad \varphi_0 = \mathrm{Id} \quad \text{where} \quad (\varphi_t)_\sharp(\rho_Z \mathrm{d}x) = \rho_t \mathrm{d}x, \tag{1}$$

then we can use $\varphi_1$ and $\rho_Z$ to sample from $\rho_D$. An important question is, whether for a given family of functions $f_t$, such velocity fields exists. While, for a fixed time $t$, the existence of such velocity fields is well established, see e.g. Laugesen et al. (2014), global existence with integrability of $v : [0, 1] \times \mathbb{R}^d \to \mathbb{R}^d$ in time and space is addressed in this paper. This is directly related to the question whether a large class of Fisher-Rao curves $\rho_t$ is absolutely continuous in the Wasserstein geometry, and in particular, fulfills a continuity equation. This can be reduced to finding solutions of a certain family of PDEs. With the appropriate Hilbert spaces at hand, we show the existence of the solutions of these PDEs and the existence of an integrable velocity field $v_t$ in time and space which admits moreover a minimality property of $\|v_t\|_{L_2(\mathbb{R}^d, \rho_t)}$.

Determining $\varphi_1$ requires both finding a curve $f_t$ and the associated vector field $v_t$. One way is to choose a curve interpolating between $f_Z$ and $f_D$ first and then to learn the velocity field. For the linear interpolation $f_t = (1 - t)f_Z + tf_D$, using above considerations, we show that there indeed exists an integrable velocity field implying the absolute continuity of the curve in the Wasserstein space. Yet, it was numerically shown that the corresponding velocity field is often badly behaved as noted in (Máté & Fleuret, 2023, Figure 4 and 6). We highlight by an analytic example the bad regularity of this flow and show in particular that its velocity field may have an exploding norm $\|v_t\|_{L_2(\mathbb{R}^d, \rho_t)}$. Another approach, recently proposed by Máté & Fleuret (2023), considers

$f_t = (1-t)f_Z + tf_D + t(1-t)\psi_t$ with unknown $\psi_t$ and learns $(\psi_t, v_t)$ simultaneously. Henceforth, we refer to this method as the "learned interpolation". However, it is theoretically unclear whether the learned curve is well-behaved or the velocity field is optimal with respect to the above norm. As an alternative approach, we propose instead to deal *directly* with a well-behaved and mathematically accessible curve on $[0,T]$ and consider the parameterization $f_t = \frac{T-t}{T}f_D + t\psi_t$. Fixing the velocity field as $v_t := \nabla(f_t - f_Z)$, we only have to learn $\psi_t$. The resulting PDE is the well-known Fokker-Planck equation related to Langevin dynamics which has a solution with Boltzmann densities $\rho_t$, if, e.g., $\rho_Z$ is a Gaussian. Then, the corresponding SDE is the Ornstein-Uhlenbeck (OU) process. Here, the backward ODE of (1) must be applied for sampling. Finally, we learn the networks for the above three cases and demonstrate by numerical examples the effectiveness of our approach.

**Contributions.**

1. We prove for the first time that under mild conditions every reasonable curve of Boltzmann densities is an absolutely continuous Wasserstein curve, see Theorems 1 and 2 and Appendix A. In particular, this includes the linear interpolation.

2. We verify by a nontrivial analytic example that the velocity field corresponding to the linear interpolation can admit an exploding norm $\|v_t\|_{L_2(\mathbb{R}^d, \rho_t)}$, leading to a "non-smooth" particle transport, see Figure 1 and Appendix B.

3. We propose to learn a curve using the parameterization $f_t = \frac{T-t}{T}f_D + t\psi_t$, which we call "gradient flow interpolation" by the following reason. With an appropriate velocity field which can be computed directly from $f_t$, this resembles the Wasserstein *gradient flow* of the Kullback-Leibler divergence with fixed second argument $\rho_Z$. This curve can be also described as the Fokker-Planck equation of a Langevin SDE.

4. We learn the networks for the linear, learned and gradient flow interpolations and demonstrate the performance of these three methods on certain sampling problems.

**Related Work.** Classical approaches for sampling from an unnormalized density are based on the Markov Chain Monte Carlo (MCMC) method, see Gilks et al. (1995), including variants as the Hamiltonian Monte Carlo (HMM) in Hoffman & Gelman (2014) or the Metropolis Adjusted Langevin Algorithm (MALA) in Girolami & Calderhead (2011). Arising problems with the correct distribution of mass among different modes lead to the use of importance sampling Neal (2001) and sequential Monte Carlo samplers Moral et al. (2006).

Newer approaches include the simulation of gradient flows of probability measures. In Chen et al. (2024) Gaussian mixtures are used to approximate the Fisher-Rao gradient flow of the Kullback-Leibler (KL) divergence. This corresponds to a time rescaling of the linear interpolation. In Nüsken (2024); Maurais & Marzouk (2024) they approximate the latter curve by kernelizing the associated Poisson equation and simulating a corresponding interacting particle system. Closely related Stein variational gradient descent Liu & Wang (2016) kernelizes the Wasserstein gradient flow of the reverse KL divergence. Recently in Chehab & Korba (2024) they consider a different path which they term the "dilation path". The associated vector field is given as the gradient of the dilation of the target density. The convolution of the density of the dilation path with (scaled) Gaussian kernels corresponds to the density of an Ornstein-Uhlenbeck process.

Recently, (stochastic) normalizing flows have been widely used to sample from unnormalized densities Noé et al. (2019); Wu et al. (2020); Hagemann et al. (2022). In Midgley et al. (2023) this problem is tackled without using a curve of probability measures by augmenting normalizing flows, which are trained using an $\alpha$-divergence loss, with annealed importance sampling. In Xu et al. (2024) the Wasserstein gradient flow of $\mathrm{KL}(\cdot, \rho_D)$ is approximated. They learn a network for approximating the corresponding JKO scheme. In a similar approach in Hertrich & Gruhlke (2024) a functional $F$ with minimum $\rho_D$ is considered. They then learn a vector field by approximating a JKO scheme for the Wasserstein gradient flow of $F$ and combine it with an importance based rejection method. They focus on $F$ being the Kullback-Leibler divergence with target distribution $\rho_D$. Using KL for sampling is a natural choice and it is shown in Chen et al. (2023) that it is the only $f$-divergence such that the Wasserstein gradient flow does not depend on the normalization constant of $\rho_D$.

Note that parametrizing the vector field using the score $\nabla \log p_t$ is common practice for sampling via SDEs. A method that generates samples and scores from trajectories of the Fokker Planck equation of certain SDEs is presented in Boffi & Vanden-Eijnden (2023). In Vargas et al. (2024)

the vector field is learned by using the KL divergence on the path measures of a forward and corresponding backward SDE assuming knowledge of $\nabla \log p_t$. In Zhang & Chen (2022) they formulate the sampling problem in terms of a stochastic control problem and explicitly include the score into the parametrization of the vector field. The stochastic control viewpoint is also used in (Berner et al., 2024, Section 3) where the KL divergence between path measures associated to a controlled and an uncontrolled SDE is used to approximate the score of e.g. an OU process. A different approach to approximating the same process using the KL on path measures is formulated in Vargas et al. (2023a). Another way of approximating a flow ending in $\rho_D$ is to consider the variance exploding diffusion SDE as done in Akhound-Sadegh et al. (2024), or to consider the time reversal of a pinned Brownian motion as in Vargas et al. (2023b); Reu et al. (2024).

The authors of Máté & Fleuret (2023) present a loss based on the continuity equation and the special form $\frac{e^{-f_t}}{Z_t}$ of Boltzmann densities. This loss allows for using fixed interpolations $f_t$ and only learning $v_t$ as well as learning $f_t$ and $v_t$ simultaneously. Using a fixed interpolation and a loss based on the continuity equation also shows up in the physics informed neural networks (PINN) literature, see e.g. Albergo & Vanden-Eijnden (2024). When finishing this paper, we became aware of the recent preprint of Sun et al. (2024) and the workshop paper Sun et al. comparing a variety of PDE based sampling methods. They included a similar approach as the one proposed here, using the same parametrization to simulate an Ornstein-Uhlenbeck SDE. However, coming from the Wasserstein perspective, we came to the conclusion that the associated gradient flow path is especially well-behaved. We simulate the corresponding *ODE* and highlight its benefits.

For the proof that many Boltzmann density curves are absolutely continuous Wasserstein curves, the main object of interest is

$$-\nabla \cdot (\rho_t \nabla s_t) = (\alpha_t - \overline{\alpha_t}) \, \rho_t, \quad t \in [0,1] \tag{2}$$

which relates Fisher-Rao curves and Wasserstein absolutely continuous curves. Here $\nabla s_t$ corresponds to the vector field of a Wasserstein absolutely continuous curve and $\alpha_t - \bar{\alpha}_t$ for $\bar{\alpha}_t = \mathbb{E}_{\rho_t}[\alpha_t]$ corresponds to the gradient in the Fisher-Rao geometry. This equation was studied for a fixed time $t$ in various contexts. In nonlinear filtering it is used to study weak solutions of nonlinear filters, see e.g. (Laugesen et al., 2014, Section 2.1). Their theory for fixed time steps $t$ can be generalized to the case of a time dependent equation as seen in Section A. The close relationship between the Poincaré inequality and fixed time solutions of (2) is discussed in Dhara & Kałamajska (2015).

## 2 WASSERSTEIN FLOWS OF BOLTZMANN DENSITIES

### 2.1 GENERAL BACKGROUND

Let $\mathcal{P}(\mathbb{R}^d)$ denote the space of probability measures on $\mathbb{R}^d$. For $\mu \in \mathcal{P}(\mathbb{R}^d)$, let $L^p(\mathbb{R}^d, \mu)$, $p \in [1, \infty)$, denotes the Banach space of (equivalence classes of) real-valued functions on $\mathbb{R}^d$ with finite norm $\|f\|_{L_p(\mathbb{R}^d,\mu)} := \left( \int_{\mathbb{R}^d} |f|^p \, \mathrm{d}\mu \right)^{1/p}$, and $L_p(\mathbb{R}^d, \mathbb{R}^d, \mu)$ be the Banach space of Borel vector fields $v : \mathbb{R}^d \to \mathbb{R}^d$ with finite norm $\|v\|_{L_p(\mathbb{R}^d,\mathbb{R}^d,\mu)} := \left( \int_{\mathbb{R}^d} \|\|v\|\|^p \, \mathrm{d}\mu \right)^{1/p}$, where $\| \cdot \|$ is the Euclidean norm on $\mathbb{R}^d$. For the Lebesgue measure, we skip the $\mu$ in the notation. The space $\mathcal{P}_2(\mathbb{R}^d)$ of probability measures having finite second moments equipped with the Wasserstein(-2) metric

$$W_2(\mu_0, \mu_1) := \min_{\pi \in \Gamma(\mu_0, \mu_1)} \left( \int_{\mathbb{R}^d \times \mathbb{R}^d} \|x - y\|^2 \, \mathrm{d}\pi(x,y) \right)^{\frac{1}{2}},$$

where $\Gamma(\mu_0, \mu_1)$ denotes the set of couplings or plans $\pi \in \mathcal{P}(\mathbb{R}^d \times \mathbb{R}^d)$ having marginals $\mu_i \in \mathcal{P}_2(\mathbb{R}^d)$, $i = 0, 1$, is a complete metric space. In complete metric spaces, we can consider absolutely continuous curves depending on the metric of the space. In the above Wasserstein space, a weakly continuous curve $\mu : [0, 1] \to \mathcal{P}_2(\mathbb{R}^d)$, $t \mapsto \mu_t$ is *absolutely continuous*, if there exists a Borel vector field $v : [0, 1] \times \mathbb{R}^d \to \mathbb{R}^d$ with $\int_0^1 \|v_t\|_{L_2(\mathbb{R}^d,\mathbb{R}^d,\mu_t)} \, \mathrm{d}t < \infty$ such that the *continuity equation*

$$\partial_t \mu_t + \nabla \cdot (\mu_t v_t) = 0 \tag{3}$$

is fulfilled in the sense of distributions [1] see, e.g. Ambrosio et al. (2005). While for fixed $\mu_t$ there are many vector fields such that the continuity equation is fulfilled, there exists a unique one such

---

[1] $\int_0^1 \int_{\mathbb{R}^d} \partial_t \phi + \langle \nabla_x \phi, v_t \rangle \, \mathrm{d}\mu_t \mathrm{d}t = 0$ for all $\phi \in C_c^\infty \left( (0,1) \times \mathbb{R}^d \right)$ where $C_c^\infty((0,1) \times \mathbb{R}^d)$ denotes the space of smooth and compactly supported functions.

that $\|v_t\|_{L_2(\mathbb{R}^d, \mu_t)}$ becomes minimal for almost every $t \in [0, 1]$. In other words, if $\mu_t$ fulfills the continuity equation for some vector field, then there exists a unique vector field that solves

$$\underset{v_t \in L_2(\mathbb{R}^d, \mathbb{R}^d, \mu_t)}{\arg \min} \int_{\mathbb{R}^d} \|v_t\|^2 \, \mathrm{d}\mu_t \quad \text{s.t.} \quad \partial_t \mu_t + \nabla \cdot (\mu_t v_t) = 0. \tag{4}$$

Moreover, this optimal vector field is determined by the condition that $v_t \in T_{\mu_t} \mathcal{P}_2(\mathbb{R}^d)$ for almost every $t \in [0, 1]$, with the *regular tangent space*

$$T_\mu \mathcal{P}_2(\mathbb{R}^d) := \overline{\{\nabla \phi : \phi \in C_c^\infty(\mathbb{R}^d)\}}^{L_2(\mathbb{R}^d, \mu)}, \tag{5}$$

where the right-hand side is the closure in the Banach space $L_2(\mathbb{R}^d, \mu)$, see (Ambrosio et al., 2005, § 8). Finally, if $\mu_t$ is an absolutely continuous curve with Borel vector field $v_t$ such that for every compact Borel set $B \subset \mathbb{R}^d$ it holds $\int_0^1 \sup_B \|v_t\| + \mathrm{Lip}(v_t, B) \, \mathrm{d}t < \infty$, then there exists a solution $\varphi : [0, 1] \times \mathbb{R}^d \to \mathbb{R}^d$ of

$$\partial_t \varphi(t, x) = v_t(\varphi(t, x)), \quad \varphi(0, x) = x \tag{6}$$

which fulfills $\mu_t = \varphi_{t, \sharp} \mu_0 := \mu_0 \circ \varphi_t^{-1}$, see e.g. (Ambrosio et al., 2005, Proposition 8.1.8).

## 2.2 WASSERSTEIN MEETS FISHER-RAO FLOWS

In the following, we are interested in absolutely continuous probability measures with respect to the Lebesgue measure, i.e., the space $\mathcal{P}^{ac}(\mathbb{R}^d) := \{\mu = \rho \, \mathrm{d}x : \rho \in L_1(\mathbb{R}^d), \rho \geq 0, \int_{\mathbb{R}^d} \rho \, \mathrm{d}x = 1\}$ and we will address such measures just by their densities. Moreover, we will only consider Boltzmann/Gibbs densities depending on functions $f : [0, 1] \times \mathbb{R}^d \to \mathbb{R}$ with $e^{-f_t} \in L_1(\mathbb{R}^d)$ and $f(\cdot, x) \in C^1([0, 1])$ for all $x \in \mathbb{R}^d$ and $\partial_t f(t, \cdot) \in L_2(\mathbb{R}^d, \rho_t)$ such that

$$\rho_t := \frac{e^{-f_t}}{Z_t}, \quad Z_t := \int_{\mathbb{R}^d} e^{-f_t} \, \mathrm{d}x. \tag{7}$$

In this case, straightforward computation gives

$$\partial_t \mu_t = \partial_t \rho_t = -(\partial_t f_t - \mathbb{E}_{\rho_t}[\partial_t f_t]) \, \rho_t, \tag{8}$$

where it maybe useful to note that $\mathbb{E}_{\rho_t}[\partial_t f_t] = -\partial_t \log Z_t$. This differential equation is a Fisher-Rao flow equation. More precisely, a weakly continuous curve $\rho : [0, 1] \to \mathcal{P}^{ac}(\mathbb{R}^d), t \mapsto \rho_t$ is a *Fisher-Rao curve*, if there exists a measurable map $\alpha : [0, 1] \times \mathbb{R}^d \to \mathbb{R}$ such that $\alpha_t := \alpha(t, \cdot) \in L_2(\mathbb{R}^d, \rho_t)$ and we have in a distibutional sense

$$\partial_t \rho_t = (\alpha_t - \bar{\alpha}_t) \rho_t, \quad \bar{\alpha}_t := \mathbb{E}_{\rho_t}[\alpha_t]. \tag{9}$$

Obviously, (8) delivers a Fisher-Rao curve with $\alpha_t := -\partial_t f_t$. For more information on Fisher-Rao flows see, e.g., Gallouët & Monsaingeon (2017); Wang & Li (2022). We are interested in the question under which conditions a Fisher-Rao curve determined by (8) is also an absolutely continuous curve in the Wasserstein space. For later purposes, note that for Boltzmann densities (7), the second summand of the continuity equation (3) reads as

$$\nabla \cdot (\mu_t v_t) = \nabla \cdot (\rho_t v_t) = (-\langle \nabla f_t, v_t \rangle + \nabla \cdot v_t) \, \rho_t. \tag{10}$$

Nevertheless, using (8), for our Fisher-Rao curves the continuity equation becomes

$$(\partial_t f_t - \mathbb{E}_{\rho_t}[\partial_t f_t]) \, \rho_t = \nabla \cdot (\rho_t v_t), \tag{11}$$

and in the ideal case that we can find a vector field $v_t = \nabla s_t$, compare (5), this can be rewritten as the family of Poisson equations

$$(\partial_t f_t - \mathbb{E}_{\rho_t}[\partial_t f_t]) \, \rho_t = \nabla \cdot (\rho_t \nabla s_t), \tag{12}$$

which needs to be solved for $s_t$. For *fixed time* $t$, the solution of the Poisson equation has already been examined, e.g., in Laugesen et al. (2014) and for more recent work, see also Reich (2011); Taghvaei & Mehta (2023). However, the pointwise solution for each fixed $t$ does not ensure that $(\rho_t, \nabla s_t)$ gives rise to an absolutely continuous Wasserstein curve, since by definition this requires additional properties of $\nabla_x s : [0, 1] \times \mathbb{R}^d \to \mathbb{R}^d$, globally in $t$: i) $\nabla_x s$ is Borel measurable on $[0, 1] \times \mathbb{R}^d$, and ii) $t \mapsto \|\nabla_x s(t, \cdot)\|_{L_2(\rho_t, \mathbb{R}^d)}^2 \in L_1([0, 1])$. If also, iii) $\nabla_x s(t, \cdot) \in T_{\rho_t} \mathcal{P}_2(\mathbb{R}^d)$ for a.e. $t \in [0, 1]$, then it is known that the velocity field $\nabla_x s$ is the *optimal one* in the sense of (4). To our best knowledge, the following two theorems, which are detailed in Appendix A, tackle a *rigorous* solution of this problem for the first time.

**Theorem 1.** *Assume that $\boldsymbol{\rho}$ determined by $\int_{[0,1]\times\mathbb{R}^d} f(t,x)\,\mathrm{d}\boldsymbol{\rho} = \int_0^1 \int_{\mathbb{R}^d} f(t,x)\,\mathrm{d}\rho_t \mathrm{d}t$ satisfies a so-called partial Poincaré inequality, see Definition A.7, for some $K > 0$, and that $\frac{1}{\boldsymbol{\rho}}$ is locally integrable. Furthermore, suppose that $\alpha_t - \overline{\alpha_t} \in L_2([0,1]\times\mathbb{R}^d, \boldsymbol{\rho})$. Then, there exists a unique weak solution $\boldsymbol{s}$ in a certain Hilbert space $\mathcal{X}_0^1$ of the problem*

$$\int_0^1 \int_{\mathbb{R}^d} \langle \nabla \boldsymbol{s}_t, \nabla \psi_t \rangle \, \mathrm{d}\rho_t \mathrm{d}t = \int_0^1 \int_{\mathbb{R}^d} \psi_t \left( \alpha_t - \overline{\alpha_t} \right) \, \mathrm{d}\rho_t \mathrm{d}t, \quad \text{for all } \psi \in \mathcal{X}_0^1. \tag{13}$$

**Theorem 2.** *Let $\rho_0, \rho_1 \in \mathcal{P}(\mathbb{R}^d)$ and assume that $\rho_t$ is a Fisher-Rao curve defined by (9), Let $\boldsymbol{\rho}$ and $\alpha$ satisfy the assumptions of Theorem 1. Then, $\rho_t$ is a Wasserstein absolutely continuous curve with vector field $\nabla_x \boldsymbol{s}$, where $\boldsymbol{s} \in \mathcal{X}_0^1$ is the unique weak solution of (13). If $\rho_t$ is bounded from above, it holds that $\nabla_x \boldsymbol{s}_t \in T_{\rho_t} \mathcal{P}_2(\mathbb{R}^d)$ for a.e. $t \in [0,1]$. Furthermore, it holds that*

$$\int_0^1 \int_{\mathbb{R}^d} \|\nabla_x \boldsymbol{s}_t\|^2 \, \mathrm{d}\rho_t \mathrm{d}t \leq K \int_0^1 \int_{\mathbb{R}^d} (\alpha_t - \overline{\alpha_t})^2 \, \mathrm{d}\rho_t \mathrm{d}t. \tag{14}$$

Note that by the Benamou-Brenier formula, see (Ambrosio et al., 2005, Equation 8.0.3), the left-hand side of (14) is an upper bound for the Wasserstein distance $W_2^2(\rho_0, \rho_1)$, while the double-integral on the right-hand side defines the Fisher-Rao *action* of $\rho_t$.

## 3 NEURAL SAMPLING FROM BOLTZMANN DENSITIES

Our aim is to sample from a Boltzmann measure $\mu_D = \rho_D \, \mathrm{d}x$, where we only have access to $f_D : \mathbb{R}^d \to \mathbb{R}$, but not to the normalization constant $Z_D$, by following a measure flow starting in a simple to sample measure $\mu_Z = \rho_Z \, \mathrm{d}x$ like the standard Gaussian one. We apply the continuity equation (11) of Fisher-Rao curves, which can be rewritten by (10) and division by $\rho_t > 0$ as

$$\partial_t f_t - \underbrace{\mathbb{E}_{\rho_t}[\partial_t f_t]}_{C_t} = -\langle \nabla f_t, v_t \rangle + \nabla \cdot v_t. \tag{15}$$

Indeed every spatially constant function $C_t$ which fulfills the above equation must be of the form $C_t = \mathbb{E}_{\rho_t}[\partial_t f_t]$, see Máté & Fleuret (2023). This enables us to learn $C_t$, so that (15) does no longer depend on normalization constants. We will deal with the following parameterizations of $f_t$ and $v_t$ and how to learn them:

1. linear interpolation: $f_t := (1-t)f_Z + tf_D \hookrightarrow$ learn $v_t$,
2. learned interpolation by Máté and Fleuret: $f_t := (1-t)f_Z + tf_D + t(1-t)\psi_t \hookrightarrow$ learn $(\psi_t, v_t)$,
3. gradient flow interpolation: $f_t := \frac{T-t}{T}f_D + t\psi_t, v_t := \nabla(f_t - f_Z) \hookrightarrow$ learn $\psi_t$.

While by definition the first two settings interpolate between $\rho_Z$ and $\rho_D$ and $\rho_0 = \rho_Z$, $\rho_1 = \rho_D$, we will see that the newly proposed third variant interpolates between $\rho_D$ and $\rho_Z$ by the choice of $v_t$, i.e., $\rho_0 = \rho_D$ and $\lim_{T\to\infty} \rho_T = \rho_Z$. In practice we reparametrize the latter curve into unit time, see Remark 4.

### 3.1 LINEAR & LEARNED INTERPOLATION

**Linear interpolation.** The simplest interpolation is the linear one $f_t := (1-t)f_Z + tf_D$, i.e., $\rho_t \propto \rho_0^{1-t} \rho_1^t$. Under suitable conditions on the end points this density fulfills the assumptions of Theorem 2, see Corollary A.17 in the Appendix, and is therefore a Wasserstein curve. By (8), it also defines a Fisher-Rao curve, and the corresponding Fisher-Rao flow equation (9) is given by

$$\partial_t \rho_t = (f_0 - f_1 - \mathbb{E}_{\rho_t}[f_0 - f_1]) \rho_t = -\left( \log(\frac{\rho_0}{\rho_1}) - \mathbb{E}_{\rho_t}[\log(\frac{\rho_0}{\rho_1})] \right) \rho_t.$$

Note that, $\rho_t$ is even a Fisher-Rao *gradient flow* of the negative log likelihood, see also Maurais & Marzouk (2024), given by $\mathcal{E}_{\mathrm{NLL}}(\rho) := -\mathbb{E}_{\rho}\left[ \log(\frac{\rho_1}{\rho_0}) \right]$.

There are a variety of approaches for approximating the vector field corresponding to the linear interpolation. In Maurais & Marzouk (2024); Nüsken (2024) a kernel-based method for solving (12) was proposed. In Máté & Fleuret (2023) they propose to solve (15) which in case of the linear interpolation becomes

$$f_1 - f_0 - C_t + \langle \nabla f_t, v_t \rangle - \nabla \cdot v_t = 0. \tag{16}$$

This equation has many solutions $v_t$ for the same $\rho_t$, and in order to enforce uniqueness we could demand that $v_t = \nabla s_t$.

This approach leads to learning neural networks $(v_t^{\theta_1}, C_t^{\theta_2})$ by minimizing the loss function

$$L(\theta) := \mathbb{E}_{t \in \mathcal{U}[0,1], x \in \mathcal{U}[a,b]^d} \left[ \mathcal{E}(\theta, x, t) \right], \tag{17}$$

$$\mathcal{E}(\theta, x, t) := |f_1 - f_0 - C_t^{\theta_2} + \langle \nabla f_t, v_t^{\theta_1} \rangle - \nabla \cdot v_t^{\theta_1}|^2, \quad \theta := (\theta_1, \theta_2).$$

**Remark 3.** *Since the approximation of high dimensional integrals is difficult Máté & Fleuret (2023) propose to use an iterative approach by solving the ODE $\partial_t \varphi_t^{\theta'} = v_t^{\theta'}(\varphi_t^{\theta'})$, $\varphi_0^{\theta'} = \mathrm{Id}$ for $\theta'$ from the previous minimization and to solve*

$$L(\theta) := \mathbb{E}_{t \in \mathcal{U}[0,1], z \sim \rho_0}[\mathcal{E}(\theta, \varphi_t^{\theta'}(z), t)].$$

*Heuristically this penalizes $\mathcal{E}(\theta, x, t)$ stronger in regions where the intermediate solution has more mass. Finally, sampling can be done – assuming sufficient regularity of the learned $v_t^{\theta_i}$ – by simulating the ODE (6).*

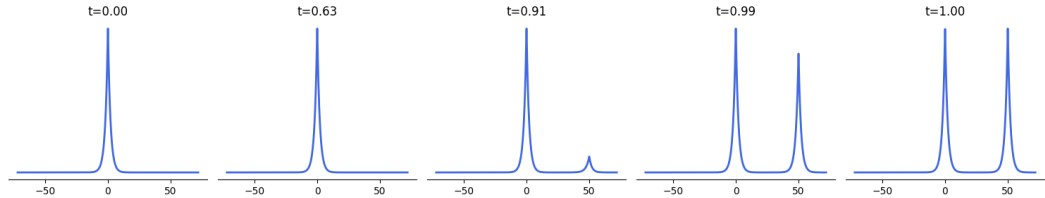

Figure 1: Evolution of the probability densities $\rho_t \propto \rho_0^{1-t} \rho_1^t$, where $\rho_0 \propto e^{-|x|}$ and $\rho_1 \propto e^{-2 \min\{|x|, |x-m|\}}$ for $m = 50$. For different values of $m$ see 5.

*A teleportation issue with the linear interpolation.* Máté & Fleuret (2023) gave examples, where for an asymmetric constellation of target and prior measure, this path transports mass into distant modes only at very late times (sometimes referred to as *mode switching*) which may cause problems while learning the corresponding vector field. Mathematically, this corresponds to the fact that the vector field $v_t(x)$ develops a singularity in $x \in \mathbb{R}^d$ for late times $t$, see (Máté & Fleuret, 2023, Figure 6). In this work, we demonstrate that this phenomenon is *more severe*: even when averaging over the space $x$, the norm $\|v_t\|_{L_2(\rho_t)}$ of the optimal field (4) explodes for late times $t$. For an example in $\mathbb{R}^1$ involving a Laplacian $\rho_0 \propto e^{-|x|}$ centered at 0 and a Laplacian-like distribution $\rho_1 \propto e^{-2 \min\{|x|, |x-m|\}}$ with a mode at 0 and a second mode at $m > 0$, see Figure 1, we are able to give an explicit and analytic formula for the calculation of $\|v_t\|_{L_2(\rho_t)}$ in the Appendix B. Using this formula (37), we demonstrate that $\|v_t\|_{L_2(\rho_t)}$ explodes when $t$ is close to 1, and that this explosion happens more severely and at later times, the further the second mode $m > 0$ is away from 0, see Figure 2.

**Learned Interpolation.** To cope with the above disadvantages of the linear interpolation, Máté & Fleuret (2023) proposed to parameterize also the flow by $f_t := (1-t)f_0 + t f_1 + t(1-t)\psi_t$ with a neural network $\psi_t^{\theta_1}$ and to minimize according to (15) the above loss with

$$\mathcal{E}(\theta, x, t) := |\partial_t f_t^{\theta_1}(x) - C_t^{\theta_3} + \langle \nabla f_t^{\theta_1}, v_t^{\theta_2} \rangle - \nabla \cdot v_t^{\theta_2}|^2, \quad \theta := (\theta_1, \theta_2, \theta_3). \tag{18}$$

and

$$L(\theta) := \mathbb{E}_{t \in \mathcal{U}[0,1], z \sim \rho_0}[\mathcal{E}(\theta, \varphi_t^{\theta'}(z), t)],$$

where $\varphi_t^{\theta'}(z)$ is as in Remark 3 a trajectory of the intermediate solution. Note that in general there are infinitely many solutions of (16) and neither the parameterized curve $\rho_t$ nor the vector field $v_t$ is unique. Further $v_t^{\theta_2}$ obtained in this way may not be minimal in any sense. The proof of the validity of the interpolation with respect to Theorem 2 is future work.

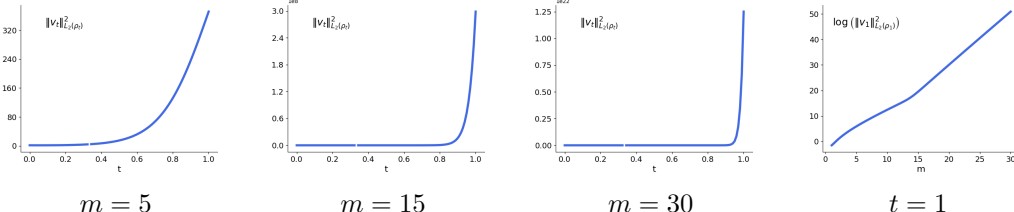

$$m = 5 \qquad\qquad m = 15 \qquad\qquad m = 30 \qquad\qquad t = 1$$

Figure 2: In the first three figures, the evolution of $\|v_t\|^2_{L_2(\rho_t)}$ belonging to the path in Figure 1 is depicted for different values of the second mode $m > 0$. For larger $m$, the norm $\|v_t\|^2_{L_2(\rho_t)}$ approaches the limit $\|v_1\|^2_{L_2(\rho_1)}$ at later times, hence with a steeper slope. The last figure shows the *log scale* of $\|v_1\|^2_{L_2(\rho_1)}$ depending on $m$. It demonstrates that $\|v_1\|_{L_2(\rho_1)}$ roughly grows *exponentially* with $m$.

## 3.2 GRADIENT FLOW INTERPOLATION

Recall that the linear interpolation of the energies $f_0, f_1$ leads to a multiplicative interpolation of densities $\rho_t(x) \propto \rho_Z(x)^{1-t}\rho_D(x)^t$. Note that this does not take account of any *local* behaviour of $\rho_Z, \rho_D$ in a neighbourhood of $x$, resulting in irregular behavior of the vector field as shown in Figure 1 and 2. In contrast, widely used paths in generative modelling Song et al. (2021); Lipman et al. (2023) originate from an interpolation of densities via (spatial) convolutions $\rho_t \sim \rho_Z(\frac{\cdot}{a_t}) * \rho_D(\frac{\cdot}{b_t})$ for time schedules $a_t, b_t$, corresponding to an interpolation $X_t = a_t Z + b_t X_0$ of associated independent random variables $Z \sim \rho_Z, X_0 \sim \rho_D$ on the *spatial level*. We aim to learn such a curve and an associated vector field.

In this paper, we propose to use $f_t := \frac{T-t}{T}f_D + t\psi_t$ and $v_t := \nabla(f_t - f_Z)$ and to parameterize $\psi_t$ by a neural network. Then, equation (15) becomes

$$\partial_t f_t - \mathbb{E}_{\rho_t}[\partial_t f_t] + \langle \nabla f_t, \nabla(f_t - f_Z)\rangle - \Delta(f_t - f_Z) = 0. \tag{19}$$

Indeed, we will show in the paragraph below that this equation makes perfect sense, in particular, it has a solution for $\rho_Z \sim \mathcal{N}(0,1)$. Then we can compute $(\psi_t^{\theta_1}, C_t^{\theta_2})$ for $f_t^{\theta_1} := \frac{T-t}{T}f_D + t\psi_t^{\theta_1}$ by minimizing the loss $L$ in (17) with the function

$$\mathcal{E}(\theta, x, t) := |\partial_t f_t^{\theta_1} - C_t^{\theta_2} + \langle \nabla f_t^{\theta_1}, \nabla(f_t^{\theta_1} - f_Z)\rangle - \Delta(f_t^{\theta_1} - f_Z)|^2, \quad \theta := (\theta_1, \theta_2). \tag{20}$$

Finally, to sample from the target distribution, we simulate the ODE (6) backwards on $[0, T]$ to obtain $\tilde{\varphi} : [0, T] \times \mathbb{R}^d \to \mathbb{R}^d$ satisfying

$$\partial_t \tilde{\varphi}(t, x) = -v_t(\tilde{\varphi}(t, x)), \quad \tilde{\varphi}(T, x) = x$$

which fulfills $\mu_t = \tilde{\varphi}_{T-t,\sharp}\mu_Z$.

**Relation to Wasserstein gradient flows.** Equation (19) is well-known in connection with Wasserstein gradient flows. A *Wasserstein gradient flow* is an absolutely continuous curve whose vector field $v_t = -\operatorname{grad}_{\rho_t} F \in T_{\rho_t}(\mathcal{P}_2(\mathbb{R}^d))$ is given by the negative Wasserstein gradient (or more generally, *subdifferential*) of a function $F : \mathcal{P}_2(\mathbb{R}^d) \to \mathbb{R}$. Then, the continuity equation

$$\partial_t \rho_t - \nabla \cdot \left(\rho_t \operatorname{grad}_{\rho_t} F\right) = 0 \tag{21}$$

produces a flow towards a minimizer of $F$. A prominent example, see, e.g. (Ambrosio et al., 2005, Chapter 10.4), is $F(\rho) := \mathrm{KL}(\rho|\rho_Z)$ arising from the Kullback-Leibler divergence which has the unique minimizer $\rho_Z$. Since it is easy to show, see (Ambrosio et al., 2005, Lemma 10.4.1), that $\operatorname{grad}_{\rho_t} F = \nabla \log \frac{\rho_t}{\rho_Z}$, we can be rewrite (21) as

$$\partial_t \rho_t = -\nabla \cdot \left(\rho_t(\nabla \log \rho_Z - \nabla \log \rho_t)\right), \quad \rho_0 = \rho_D, \tag{22}$$

which is also known as the Fokker-Planck equation. Inserting $\rho_t := e^{-f_t}/Z_t$ results exactly in (19). The existence and uniqueness of a solution of (22) is shown in (Ambrosio & Savaré, 2007, Theorem 6.6). However, it is not clear if Boltzmann densities stay Boltzmann densities, i.e. if the densities $\rho_t$ are strictly positive for all $t$. But in the special case, when $\rho_Z$ is a Gaussian, this is the case as we

will see next.

There is a close relation between (22) and SDEs, namely the Langevin SDE

$$\mathrm{d}X_t = \nabla \log \rho_Z(X_t)\mathrm{d}t + \sqrt{2}\mathrm{d}B_t \tag{23}$$

corresponds to the Fokker-Planck equation (22), see e.g., (Chewi et al., 2024, Chapter 6.2), i.e., $X_t \sim \rho_t$. For $\rho_Z \propto \mathcal{N}(0, I_d)$, (23) becomes $\mathrm{d}X_t = -X_t\mathrm{d}t + \sqrt{2}\mathrm{d}B_t$ which is an Ornstein-Uhlenbeck (OU) process, and a minimizer of (20) is (up to a constant) $\log p_t$ of the corresponding Fokker-Planck equation. In (Sun et al., 2024, Equation (18)) the loss (20) is derived from this process. This Ornstein-Uhlenbeck process has the closed form solution

$$X_t = e^{-t}Z + e^{-t}X_0, \quad Z \sim \mathcal{N}\left(0, (e^{2t}-1)I_d\right), \quad X_0 \sim \rho_D,$$

where $Z$ is independent from $X_0$, see (Herrmann & Massin, 2019, Eq. (2.4)). From the closed form solution it follows that $\rho_t \sim \mathcal{N}\left(0, (1-e^{-2t})I_d\right) * \rho_D\left(e^t\cdot\right) > 0$, and thus $\rho_t \propto e^{-f_t}$ which implies that (19) has a solution.

**Remark 4.** *To obtain dynamics on the unit interval for the gradient flow interpolation we apply an approach from the score based diffusion literature Song et al. (2021). We use a linear time schedule $\beta(t) := \beta_{min} + t\left(\beta_{max} - \beta_{min}\right)$ and the associated SDE*

$$\mathrm{d}X_t = -\frac{1}{2}\beta(t)X_t\mathrm{d}t + \sqrt{\beta(t)}\mathrm{d}B_t \quad X_0 \sim \rho_1. \tag{24}$$

*The SDE* (24) *has a closed form solution given by*

$$X_t = \sqrt{1 - e^{-g(t)}}Z + e^{\frac{-g(t)}{2}}X_0, \tag{25}$$

*where $g(t) := \int_0^t \beta(s)\,\mathrm{d}s$ and $Z \sim \mathcal{N}(0, I_d)$. As done in Song et al. (2021) we choose $\beta_{min} = 0.1$ and $\beta_{max} = 20$. These choices ensure that the distribution of $X_1$ is approximately $\mathcal{N}(0, I_d)$. Note that in contrast, finite-time interpolations as the linear one do not incur such a mixing error. We then aim to learn the solution of the associated Fokker-Planck equation, this corresponds to setting $v_t := \frac{\beta(t)}{2}\nabla(f_t - \frac{\|\cdot\|^2}{2}).$*

The following example illustrates the behavior of the three interpolations, when learning the corresponding vector fields.

**Example 5.** *Motivated by an example in Máté & Fleuret (2023) and incorporating the idea of our teleportation example with asymmetrically diverging modes, we choose a standard Gaussian source distribution and the target distribution*

$$\frac{1}{3}\mathcal{N}\left([4,4],1\right) + \frac{2}{3}\mathcal{N}\left([-m,-m],1\right).$$

*For the linear and learned interpolations, we consider both sampling from uniform domains and sampling from the trajectories as described in Remark 3. For the gradient flow interpolation, we sample points uniformly and interpolate them with samples from our latent distribution according to formula (25). The results are reported in Figure 3. Note that, in case of the learned interpolation, the mode collapse could be eventually alleviated by choosing a different source distribution with a larger support and/or heavier tails.*

## 4 EXPERIMENTS

In this section we apply the different approaches to common sampling problems. We compare the performance of using the linear, learned or gradient flow interpolation.

For both the linear interpolation and the learned interpolation, we can choose to learn the vector field directly, or learn a potential and obtain the vector field as its gradient. In practice, we choose to parameterize it directly as in Máté & Fleuret (2023). Note that in practice for the gradient flow interpolation we apply Remark 4. Furthermore we improve performance by using learned scheduling function $g(t)$, satisfying $g(0) = 1$ and $g(1) = 0$, and parametrizing the corresponding $f_t$ as $f_t := g(t)f_D + t\psi_t$.

A major design choice is at which points to evaluate the pointwise error $\mathcal{E}(\theta, x, t)$. For the gradient flow interpolation, we chose the sampling strategy described in Example 5. For the linear and learned

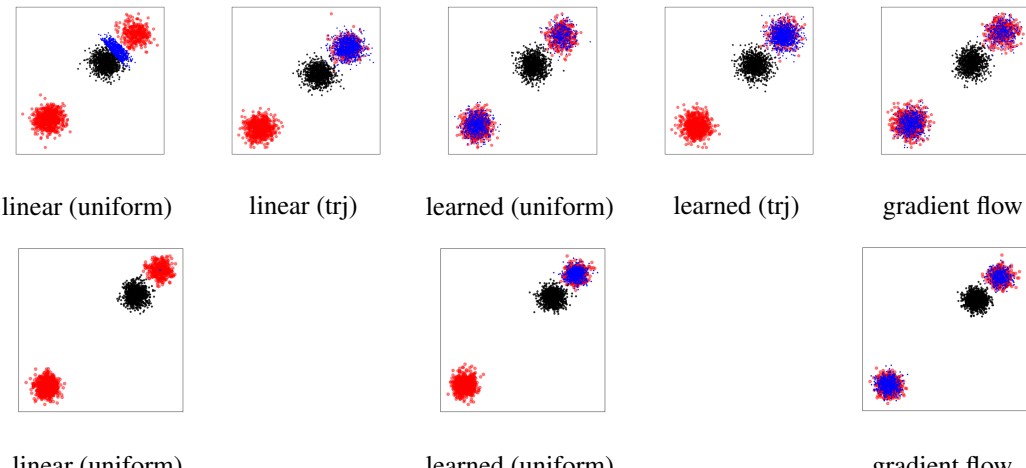

linear (uniform)    linear (trj)    learned (uniform)    learned (trj)    gradient flow

linear (uniform)    learned (uniform)    gradient flow

Figure 3: Sampling results for $m = 8$ (top) and $m = 15$ (bottom) as in Example 5: source (black), target (red) and estimated (blue). We cannot learn a meaningful vector field for the linear interpolation for both sampling strategies. Sampling along the trajectories ("trj") leads to mode collapse for the learned interpolation. While the learned interpolation with uniform sampling covers the modes for $m = 8$, increasing $m$ leads to mode collapse. In contrast, the gradient flow interpolation does not mode collapse for both $m$.

interpolation we experimented both with sampling from uniform domains, or sampling along the trajectory as in Máté & Fleuret (2023). We found in our experiments that the method employed in Máté & Fleuret (2023) led to better performance.

In the following, we evaluate the methods using the effective sample size, negative log likelihood and the energy distance Székely & Rizzo (2013). See D for further details on the implementation and evaluation.

**Gaussian Mixture Model.**    We use the experimental setup from Midgley et al. (2023). The target distribution consists of a mixture of 40 evenly weighted Gaussians in 2 dimensions. The means are distributed uniformly over $[-40, 40]^2$. In Figure 4 we plot the countour lines of the target distribu-

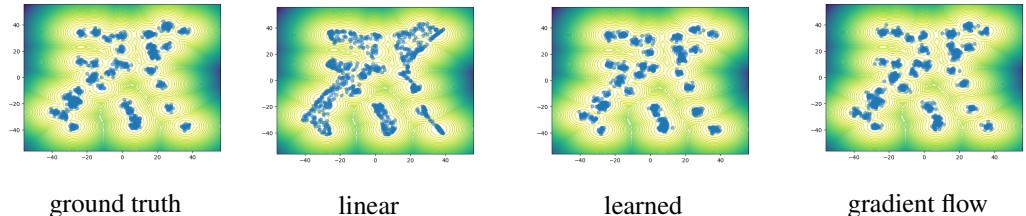

ground truth    linear    learned    gradient flow

Figure 4: Results for a Gaussian Mixture Model with 40 modes. For the linear and learned interpolation we showed the results for $\sigma = 30$, for the gradient flow interpretation we used $\sigma = 1$.

tion as well as 1000 samples from the simulated flow and the ground truth, respectively. For this experiment an important factor is the choice of latent distribution. We restricted ourselves to using a Gaussian latent distribution. However as outlined in Section 3.1 the linear interpolation faces difficulties when some modes are far away from the latent while others are close. In order to mitigate this problem, we experimented with using varying standard deviations $\sigma$; the impact of this choice is visualized in D. In contrast, the gradient flow interpolation works well when starting in a standard Gaussian which is what we do.

**Many Well Distribution.**    As target distributions we consider a 8 and 16-dimensional many well distribution similarly to the implementation of the target energy from Midgley et al. (2023). More

|  | ESS $\uparrow$ | NLL $\downarrow$ | Energy $\downarrow$ |
|---|---|---|---|
| linear ($\sigma = 30$) | $0.183_{\pm 0.05}$ | $8.657_{\pm 0.012}$ | $0.247_{\pm 0.016}$ |
| learned ($\sigma = 20$) | $0.605_{\pm 0.316}$ | $6.894_{\pm 0.005}$ | $0.058_{\pm 0.006}$ |
| learned ($\sigma = 30$) | $0.922_{\pm 0.002}$ | $6.913_{\pm 0.003}$ | $0.121_{\pm 0.009}$ |
| gradient flow ($\sigma = 1$) | $0.992_{\pm 5e\text{-}5}$ | $6.864_{\pm 0.005}$ | $0.0021_{\pm 0.001}$ |

Table 1: Comparison of effective sample size (ESS), negative log likelihood (NLL), and the energy distance for different interpolations. Arrows indicate if ascending/descending values are better.

precisely, let

$$\log p_{m,d}(x) = \sum_{i=1}^{m} \left( -x_i^4 + 6x_i^2 + \frac{1}{2}x_i \right) + \sum_{i=m+1}^{d} -\frac{1}{2}x_i^2 + \text{constant}.$$

The term $-x_i^4 + 6x_i^2 + \frac{1}{2}x_i$ creates two modes with unequal weights and the number of overall modes of $p_{m,d}$ is $2^m$. Then our 8 dimensional many well distribution is $p_{4,8}$ and the 16 dimensional target distribution is $p_{4,16}$. The results in Table 2 show that both the learned and gradient flow interpolations significantly outperform the linear interpolation for both dimensions. While the learned and gradient flow interpolations are competitive in $d = 8$, the learned interpolation works better in $d = 16$. The latter behavior might be due to greater flexibility, since both the curve and the vector field are parametrized separately. Here the choice of latent distribution is not as relevant as starting in a standard Gaussian is sufficient for all methods, since the modes of the many well distribution are concentrated close to zero.

| Interpolation | 8-dimensional - $p_{4,8}$ | | | 16-dimensional - $p_{4,16}$ | | |
|---|---|---|---|---|---|---|
| | ESS $\uparrow$ | NLL $\downarrow$ | Energy $\downarrow$ | ESS $\uparrow$ | NLL $\downarrow$ | Energy $\downarrow$ |
| Linear | $0.985_{\pm 0.016}$ | $9.689_{\pm 0.01}$ | $0.278_{\pm 0.003}$ | $0.998_{\pm 3e\text{-}5}$ | $21.85_{\pm 0.01}$ | $0.201_{\pm 0.002}$ |
| Learned | $0.999_{\pm 1e\text{-}5}$ | $6.876_{\pm 0.01}$ | $9e\text{-}5_{\pm 3e\text{-}5}$ | $0.998_{\pm 3e\text{-}5}$ | $18.24_{\pm 0.01}$ | $1e\text{-}4_{\pm 3e\text{-}5}$ |
| Gradient Flow | $0.991_{\pm 3e\text{-}4}$ | $6.989_{\pm 0.01}$ | $8e\text{-}4_{\pm 1e\text{-}4}$ | $0.544_{\pm 0.005}$ | $20.53_{\pm 0.17}$ | $0.009_{\pm 4e\text{-}4}$ |

Table 2: Comparison of effective sample size (ESS), negative log likelihood (NLL), and energy distance for different interpolations, evaluated for the 8-dimensional and 16-dimensional experiments with $m = 4$. Note that the ESS is only meaningful in relation with the Energy.

## 5 CONCLUSIONS

**Discussion.** We studied different paths in the probability space in order to sample from unnormalized densities and compared their behaviour and computibility. The linear interpolation of energies suffers from an exploding vector field, and learning the vector field *and* energy path simultaneously (*learned interpolation*) in practice solves this problem. However, there is no theoretical knowledge about the path which we want to approximate and in turn no conclusions about the regularity of the corresponding vector field can be drawn. In contrast, the proposed gradient flow approach allows for a thorough mathematical description of the velocity field and the path itself, opening the doors for a rigorous analysis via Wasserstein gradient flows and Fokker-Planck equations. We show numerical performance on sampling from Gaussian mixture models and a many well distribution. Furthermore we close a gap in the literature by showing that a large class of Fisher Rao curves and in particular curves of Boltzmann densities are in fact absolutely continuous curves in the Wasserstein geometry. Hence the existence of a corresponding vector field driving the evolution is implied, justifying sampling from the target density via solving an ODE.

**Limitations and Outlook.** The methods proposed require knowledge of the domain of the target distribution. Furthermore computation of divergences and Laplacians is expensive in high dimensions, requiring the use of e.g. Hutchinsons trace estimator. These issues can be addressed by developing more efficient strategies to sample points at which to evaluate the loss $\mathcal{E}(\theta, x, t)$, such as combining uniform sampling and trajectory based sampling. From a theoretical point of view further investigation of the behaviour and regularity of the gradient flow path is needed, especially the behaviour of the norm $\|v_t\|_{L_2(\rho_t)}$ would be of interest.

ACKNOWLEDGMENTS

J.C. acknowledges funding by the German Research Foundation (DFG) within the project SPP 2298 "Theoretical Foundations of Deep Learning", C.W. acknowledges funding by the DFG within the SFB "Tomography Across the Scales" (STE 571/19-1, project number: 495365311), R.D. acknowledges funding by the German Research Foundation (DFG) within the project STE 571/16-1.

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

## A  WASSERSTEIN MEETS FISHER-RAO: SOLUTION OF A FAMILY OF POISSON EQUATIONS

We will show that under mild assumptions on $f$, the Fisher-Rao curve $\rho_t$ from (8) is also a Wasserstein absolutely continuous curve with a tangent vector field $\nabla s_t$[2]. In order to do that, we will

---

[2]Overall, let $\nabla := \nabla_x$ denote differentiation in direction $x \in \mathbb{R}^d$.

describe $s_t$ as the unique solution of a Poisson equation in a suitable Hilbert space $\mathcal{X}_0^1$. More precisely, we show that there exists a unique $s \in L_2([0,1] \times \mathbb{R}^d; \boldsymbol{\rho})$ (with $\boldsymbol{\rho}$ given by Definition A.1) such that $\nabla s_t := \nabla s(t, \cdot)$ fulfills the Poisson equation

$$\nabla \cdot (\rho_t \nabla s_t) = \left( \partial_t f - \mathbb{E}_{\rho_t} [\partial_t f] \right) \rho_t \tag{26}$$

in a distributional sense in $\mathcal{X}_0^1$. For fixed time $t$, this has already been done, e.g., in Laugesen et al. (2014) and Vargas et al. (2024). But note that a pointwise-in-time treatment does not allow us to conclude *Wasserstein absolute continuity* of our curve $\rho_t$. Therefore, we generalize their methods to a *global-in-time* ansatz; see also the discussion at the end of Section A. In particular, under suitable assumptions on $f$, we find a unique weak solution $s$ of (26) such that

1. $\nabla_x s$ is Borel measurable on $[0,1] \times \mathbb{R}^d$,

2. $t \mapsto \|\nabla_x s(t, \cdot)\|_{L_2(\rho_t)}^2 \in L_1([0,1])$, i.e., $\int_0^1 \|\nabla_x s(t, \cdot)\|_{L_2(\rho_t)}^2 \, \mathrm{d}t < \infty$,

3. $\nabla_x s(t, \cdot) \in T_{\rho_t} \mathcal{P}_2(\mathbb{R}^d)$ for a.e. $t \in [0,1]$.

These conditions ensure that the Fisher-Rao curve (8) is absolutely continuous in the Wasserstein sense, and that the velocity field $\nabla_x s$ is the *optimal one* fulfilling the continuity equation w.r.t. our curve $\rho_t$ as described in (4).

As a by-product, we obtain the estimate

$$\int_0^1 \int_{\mathbb{R}^d} \|\nabla_x s(t, \cdot)\|^2 \, \mathrm{d}\rho_t \mathrm{d}t \leq K \int_0^1 \int_{\mathbb{R}^d} \left( \alpha_t - \overline{\alpha_t} \right)^2 \, \mathrm{d}\rho_t \mathrm{d}t,$$

linking (an upper bound of) the Wasserstein-2 distance $W_2^2(\rho_0, \rho_1)$ on the one side, and the Fisher-Rao action on the other.

Next we will define the appropriate spaces for our analysis.

### A.1 PREREQUISITES FOR SOLVING THE POISSON EQUATION

**Definition A.1.** *Let $t \mapsto \rho_t$ be a weakly continuous curve $[0,1] \to \mathcal{P}^{\mathrm{ac}}(\mathbb{R}^d)$. Then, for all Borel measurable $B \subseteq \mathbb{R}^d$, the map $t \mapsto \rho_t(B)$ is measurable: Indeed, it is the pointwise limit of continuous, hence measurable maps $t \mapsto \int_{\mathbb{R}^d} f_k \, \mathrm{d}\rho_t$ with appropriate $f_k$. Thus, $\boldsymbol{\rho}$ defined by the disintegration $\int_{[0,1] \times \mathbb{R}^d} f(t, x) \, \mathrm{d}\boldsymbol{\rho} = \int_0^1 \int_{\mathbb{R}^d} f(t, x) \, \mathrm{d}\rho_t \mathrm{d}t$, is a probability measure on $[0,1] \times \mathbb{R}^d$, see e.g., (Ambrosio et al., 2005, Chapter 5.3).*

**Definition A.2.** *Let $\rho \, \mathrm{d}x \in \mathcal{P}^{\mathrm{ac}}(\mathbb{R}^d)$ with density $\rho$. We consider the Sobolev space*

$$H^1(\rho) := \Big\{ f \in L_2(\mathbb{R}^d, \rho) : \text{the weak derivative } \partial_{x_i} f \text{ (w.r.t. } \mathcal{L}) \text{ exists and}$$

$$\partial_{x_i} f \in L_2(\rho), \, \forall i \in \{1, \ldots, d\} \Big\}$$

*with inner product $\langle f, g \rangle_{H^1(\rho)} := \langle f, g \rangle_{L_2(\rho)} + \langle \nabla f, \nabla g \rangle_{L_2(\rho, \mathbb{R}^d)}$. Furthermore, define the space of Sobolev functions with zero mean by*

$$H_0^1(\rho) := \left\{ f \in H^1(\rho) : \int_{\mathbb{R}^d} f \, \mathrm{d}\rho = 0 \right\}.$$

**Lemma A.3.** *Assume that $\rho \, \mathrm{d}x \in \mathcal{P}^{\mathrm{ac}}(\mathbb{R}^d)$ and $\frac{1}{\rho} \in L_{1, \mathrm{loc}}(\mathbb{R}^d)$. Then, $(H^1(\rho), \langle \cdot, \cdot \rangle_{H^1(\rho)})$ is a Hilbert space and $H_0^1(\rho)$ is a closed subspace.*

*Proof.* The first claim is (Kufner & Opic, 1984, Theorem 1.11). For the second claim note that for $f_n \to f$ in $H^1(\rho)$ with $f_n \in H_0^1(\rho)$, we have that

$$\left| \int_{\mathbb{R}^d} f \, \mathrm{d}\rho \right| = |\langle f, 1 \rangle_{L_2(\rho)}| \leq |\langle f - f_n, 1 \rangle_{L_2(\rho)}| + |\langle f_n, 1 \rangle_{L_2(\rho)}|$$

$$\leq \|f_n - f\|_{H^1(\rho)} + 0 \to 0.$$

$\square$

**Definition A.4.** *Consider a weakly continuous curve $[0,1] \to \mathcal{P}^{\mathrm{ac}}(\mathbb{R}^d)$, $t \mapsto \rho_t$, and the associated measure $\boldsymbol{\rho} \in \mathcal{P}^{\mathrm{ac}}([0,1] \times \mathbb{R}^d)$ given by Definition A.1. Define*

$$H_x^1(\boldsymbol{\rho}) := \Big\{ f \in L_2([0,1] \times \mathbb{R}^d, \boldsymbol{\rho}) : \text{ the weak derivative } \partial_{x_i} f \text{ exists and}$$

$$\partial_{x_i} f \in L_2(\boldsymbol{\rho}), \ \forall i \in \{2, \dots, d+1\} \Big\}$$

*with respect to the inner product $\langle f, g \rangle_{H_x^1(\boldsymbol{\rho})} := \langle f, g \rangle_{L_2(\boldsymbol{\rho})} + \langle \nabla_x f, \nabla_x g \rangle_{L_2(\boldsymbol{\rho})}$, where $f = f(t,x)$, $g = g(t,x)$. Now, consider the subspace*

$$\mathcal{X}_0^1 := \Big\{ s \in H_x^1(\boldsymbol{\rho}) : s(t, \cdot) \in H_0^1(\rho_t) \text{ for almost every } t \in [0,1] \Big\}.$$

*For $s \in \mathcal{X}_0^1$ we will also write $s_t$ for denoting the function $s(t, \cdot)$.*

**Lemma A.5.** *Assume that $\frac{1}{\boldsymbol{\rho}} \in L_{1,\mathrm{loc}}([0,1] \times \mathbb{R}^d)$. Then, the space $H_x^1(\boldsymbol{\rho})$ is a Hilbert space, and $\mathcal{X}_0^1$ is a closed subspace of $H_x^1(\boldsymbol{\rho})$.*

*Proof.* The Hilbert space property of $H_x^1(\boldsymbol{\rho})$ again follows by (Kufner & Opic, 1984, Theorem 1.11) and we only prove the closedness of $\mathcal{X}_0^1$. Now, let $(s_n) \subset \mathcal{X}_0^1$ be a sequence with $s_n \to s$ in $H_x^1(\boldsymbol{\rho})$. Then,

$$\int_0^1 \int_{\mathbb{R}^d} |s_n(t, \cdot) - s(t, \cdot)|^2 + \|\nabla_x s_n(t, \cdot) - \nabla_x s(t, \cdot)\|^2 \, \mathrm{d}\rho_t \mathrm{d}t \to 0.$$

This implies pointwise convergence for a.e. $t \in [0,1]$ (along a subsequence), i.e.,

$$\int_{\mathbb{R}^d} |s_n(t, \cdot) - s(t, \cdot)|^2 + \|\nabla_x s_n(t, \cdot) - \nabla_x s(t, \cdot)\|^2 \, \mathrm{d}\rho_t \to 0 \quad \text{for a.e. } t \in [0,1].$$

Thus, we have that $s_n(t, \cdot) \to s(t, \cdot)$ converges in $H^1(\rho_t)$ for almost all $t \in [0,1]$. Since $H_0^1(\rho_t)$ is closed, we have that the limit $s(t, \cdot)$ is in $H_0^1(\rho_t)$ for a.e. $t$. $\qquad \square$

**Definition A.6.** *On $\mathcal{X}_0^1$ we define an bilinear form by setting*

$$\langle f, g \rangle_{\mathcal{X}_0^1} = \langle \nabla_x f, \nabla_x g \rangle_{L_2(\boldsymbol{\rho})}.$$

**Definition A.7.** *We say that a probability measure $\boldsymbol{\rho}$ fulfills the partial Poincaré inequality (PPI) if there exists $K > 0$ such that*

$$\|g\|_{L_2(\boldsymbol{\rho})}^2 \leq K \|\nabla_x g\|_{L_2(\boldsymbol{\rho})}^2 \tag{PPI}$$

*for all $g \in \mathcal{X}_0^1$. Note that usually it is formulated as $Var_\mu[g] \leq K \|\nabla_x g\|_{L_2(\mu)}^2$, but since we have $\mathbb{E}_{\rho_t}[g_t] = 0$ for $g \in \mathcal{X}_0^1$ both notions coincide.*

Note that the above definition is more general than the pointwise Poincaré inequality used in Vargas et al. (2024), since we can allow single time points $t$, where the inequality may fail.

**Lemma A.8.** *Assume that $\boldsymbol{\rho}$ fulfills the (PPI), and that $\frac{1}{\boldsymbol{\rho}} \in L_{1,\mathrm{loc}}([0,1] \times \mathbb{R}^d)$. Then, $(\mathcal{X}_0^1, \langle \cdot, \cdot \rangle_{\mathcal{X}_0^1})$ is a Hilbert space.*

*Proof.* We have to show that $\langle \cdot, \cdot \rangle_{\mathcal{X}_0^1}$ is positive definite and $\mathcal{X}_0^1$ is complete. The first claim follows directly from the (PPI). The second follows from the fact that $\| \cdot \|_{H_x^1(\boldsymbol{\rho})} \leq \sqrt{K+1} \| \cdot \|_{\mathcal{X}_0^1} \leq \sqrt{K+1} \| \cdot \|_{H_x^1(\boldsymbol{\rho})}$ and $\mathcal{X}_0^1$ is closed in $H_x^1(\boldsymbol{\rho})$. $\qquad \square$

This assumption (PPI) is usually fulfilled in our setting as we discuss in the following.

**Lemma A.9.** *(Bakry et al., 2008, Corollary 1.6) Let $\mu = e^{-V(x)} \, \mathrm{d}x$ be a probability measure on $\mathbb{R}^d$ and assume that $V \in C^2(\mathbb{R}^d)$ such that $V$ is lower bounded. Assume furthermore that there exists $\eta > 0$, $\tilde{R} \geq 0$ such that for all $\|x\| > \tilde{R}$ we have*

$$\langle x, \nabla V(x) \rangle \geq \eta \|x\|.$$

*Then, there exists a constant $C = C\big(d, \eta, \|V\|_{L^\infty(B(R))}, \|\nabla V\|_{L^\infty(B(R))}\big) > 0$, where $R := \max\{\frac{2d}{\eta}, \tilde{R}\}$, such that for all $\phi \in H_0^1(\mu)$, it holds*

$$\int_{\mathbb{R}^d} \phi^2(x)\, \mathrm{d}\mu \le C \int_{\mathbb{R}^d} \|\nabla \phi\|^2 \, \mathrm{d}\mu.$$

*The constant $C$ can be estimated as follows: Let $\beta := \beta_{V,R} := 2\|V\|_{L^\infty(B(R))}$. There exist $C_{1,\eta,R}, C_{2,\eta,R}, C_{3,\eta,R} > 0$ (not depending on $V$) and a universal constant $D > 0$ such that*

$$C \le \frac{4d}{\eta^2}\Big(1 + \big(C_{1,\eta,R} + C_{2,\eta,R}\|\nabla V\|_{L^\infty(B(R))} + C_{3,\eta,R}\big) D\, R^2 e^\beta\Big).$$

*Proof.* We first show the claim for all smooth $\phi \in C^\infty(\mathbb{R}^d) \cap H^1(\mu)$ with zero mean. Unfortunately in (Bakry et al., 2008, Corollary 1.6), the constants are not explicitly computed which is why we track them here. They consider for $\gamma = \eta/2$ the function given by $W(x) = W_{\eta,R}(x) := e^{\gamma\|x\|}$ for $\|x\| > R$, and smooth continuation for $\|x\| \le R$. They show that, *if* for $L := \Delta - \langle \nabla V, \nabla \cdot \rangle$, it holds that

$$LW \le -\theta W(x) + b\chi_{B(R)},$$

*then*, the inequality $Var_\mu[\phi] \le C \int_{\mathbb{R}^d} \|\nabla \phi^2\|\, \mathrm{d}\mu$ is true with constant $C = \frac{1}{\theta}(1 + b\kappa_R)$, where the constant $\kappa_R$ can be estimated by $D\, R^2 e^\beta$. We are left to upper bound the constants $\theta$ and $b$. Let $\|x\| \le R$. Then, we have that

$$|LW(x)| \le |\Delta W(x)| + |\langle \nabla V(x), \nabla W(x)\rangle| \le \|\Delta W\|_{L^\infty(B(R))} + \|\nabla V\|_{L^\infty(B(R))}\|\nabla W\|_{L^\infty(B(R))},$$

and for $\|x\| > R$, by choice of $R$,

$$LW(x) = \frac{\eta}{2}\left(\frac{d-1}{\|x\|} + \frac{\eta}{2} - \frac{\langle x, \nabla V\rangle}{\|x\|}\right) W(x) \le \frac{\eta}{2}\left(\frac{d-1}{R} + \frac{\eta}{2} - \eta\right) W(x)$$

$$\le \frac{\eta}{2}\left(\frac{\eta(d-1)}{2d} - \frac{\eta d}{2d}\right) W(x) = -\frac{\eta^2}{4d} W(x),$$

and thus the claim is true for $\phi \in C^\infty(\mathbb{R}^d) \cap H^1(\mu)$ with zero mean, using the constants $\theta = \frac{\eta^2}{4d} > 0$ and $b = C_{1,\eta,R} + C_{2,\eta,R}\|\nabla V\|_{L^\infty(B(R))} + C_{3,\eta,R}$, where $C_{1,\eta,R} := \|\Delta W_{\eta,R}\|_{L^\infty(B(R))}$, $C_{2,\eta,R} := \|\nabla W_{\eta,R}\|_{L^\infty(B(R))}$ and $C_{3,\eta,R} := \theta\|W_{\eta,R}\|_{L^\infty(B(R))}$.

In order to extend this result to $\phi \in H_0^1(\mu)$, note that $C_c^\infty(\mathbb{R}^d)$ is dense in $H^1(\mathbb{R}^d)$, see (Adams & Fournier, 2003, Cor. 3.23). Since by assumption, $V$ is lower bounded, i.e., $e^{-V}$ is upper bounded, $C_c^\infty(\mathbb{R}^d)$ is also dense in $H^1(\mu)$ by Lebesgue's dominated convergence theorem. Now take a sequence $(\phi_n) \in C_c^\infty(\mathbb{R}^d)$ converging to $\phi$ in $H^1(\mu)$. Then, also $\tilde{\phi}_n := \phi_n - \int_{\mathbb{R}^d} \phi_n \, \mathrm{d}\mu$ converges to $\phi$ in $H^1(\mu)$, and $\tilde{\phi}_n \in C^\infty(\mathbb{R}^d) \cap H^1(\mu)$ has zero mean. The first part now yields the claim. $\qquad\square$

For the specific path $\rho_t$ given by the *linear* potential interpolation $V_t = (1-t)f_0 + tf_1$, the disintegration measure $\boldsymbol{\rho}$ fulfills the partial Poincaré inequality (PPI) given suitable assumptions on the end points $f_0, f_1$, as the next proposition shows.

**Proposition A.10.** *Let $f_0, f_1 \in C^2(\mathbb{R}^d)$ be lower bounded such that there exists $\alpha, \eta > 0$ and $\tilde{R} > 0$ such that for all $\|x\| \ge \tilde{R}$ it holds the* drift condition

$$\langle x, \nabla f_i(x)\rangle \ge \eta\|x\|, \quad i = 0, 1, \tag{27}$$

*and the* linear growth condition

$$f_i(x) \ge \alpha\|x\|, \quad i = 0, 1. \tag{28}$$

*Let $V_t = (1-t)f_0 + tf_1$ and $\rho_t = Z_t^{-1} e^{-V_t}$ for $Z_t = \int_{\mathbb{R}^d} e^{-V_t}\, \mathrm{d}x$, and $\boldsymbol{\rho}$ as in Definition A.1. Then, $\boldsymbol{\rho}$ satisfies the partial Poincaré inequality (PPI), i.e., there exists $C > 0$ such that for all $\phi \in \mathcal{X}_0^1$ it holds*

$$\int_{[0,1]\times\mathbb{R}^d} \phi^2\, \mathrm{d}\boldsymbol{\rho} \le C \int_{[0,1]\times\mathbb{R}^d} \|\nabla_x \phi\|^2\, \mathrm{d}\boldsymbol{\rho}.$$

*Proof.* First note that by the lower boundedness and linear growth assumption (28) on $f_0, f_1$, the normalization constant $Z_t$ is well-defined, and $\rho_t$ is indeed a probability measure for all $t \in [0, 1]$. Further note that $\boldsymbol{\rho}$ exists if $\rho_t$ is weakly continuous, see again Definition A.1. We now show the weak continuity of $\rho_t$:

Let $g$ be a bounded and continuous function on $\mathbb{R}^d$. By the linear growth (28) of $f_0, f_1$ and dominated convergence, the map $t \mapsto Z_t = \int_{\mathbb{R}^d} e^{-V_t} \, dx$ is continuous on $[0, 1]$. Since it is also strictly positive on $[0, 1]$, also its inverse $t \mapsto Z_t^{-1}$ is continuous on $[0, 1]$. By the same token, the map $t \mapsto \int_{\mathbb{R}^d} g e^{-V_t} \, dx$ is continuous, and altogether, $t \mapsto \rho_t$ is weakly continuous.

In order to prove the inequality, we conclude by (27) that $\langle x, \nabla V_t(x) \rangle \geq (1 - t)\eta \|x\| + t\eta \|x\| \geq \eta \|x\|$ for all $\|x\| \geq \tilde{R}$. Let $R := \max\{\frac{2d}{\eta}, \tilde{R}\}$ and note that for $V(t, x) := V_t(x) + \ln(Z_t)$, it is now easy to check that $V(t, \cdot) \in C^2(\mathbb{R}^d)$, $\|\nabla_x V\|_{L^\infty([0,1] \times B(R))} < \infty$ and $\beta := 2\|V\|_{L^\infty([0,1] \times B(R))} < \infty$. Then, by Proposition A.9 we know that

$$\int_{\mathbb{R}^d} \phi(t, \cdot)^2 \, d\rho_t \leq C \int_{\mathbb{R}^d} \|\nabla_x \phi(t, \cdot)\|^2 \, d\rho_t$$

for all $\phi \in \mathcal{X}_0^1$ and all $t \in [0, 1]$, with a constant $C$ which does *not* depend on $t$, hence the claim. $\quad\square$

**Remark A.11.** *(i) Gaussian distributions given by $f_i := \frac{1}{2\sigma^2} \|x - m\|^2$ satisfy the assumptions of Proposition A.10.*

*(ii) In Proposition A.10, the regularity assumptions on the end points, i.e., $f_0, f_1 \in C^2(\mathbb{R}^d)$ can be relaxed. It is enough to assume that there exist sequences of functions $(f_{0,n}), (f_{1,n}) \subset C^2(\mathbb{R}^d)$, which converge pointwise to $f_0$ and $f_1$, respectively, such that the assumptions of Proposition A.10 are satisfied uniformly in $n$ and such that $\|\nabla_x V_n\|_{L^\infty}, \|V_n\|_{L^\infty}$ are bounded in $n$.*

*Hereby, also the Fisher-Rao curve given by linear interpolation of $f_0 := \|x\|$ and $f_1 := \min\{\|x\|, \|x - m\|\}$ with $m \in \mathbb{R}^d$, i.e., the interpolation between two Laplacian (type) distributions, satisfies* (PPI).

## A.2 EXISTENCE OF A SOLUTION TO THE POISSON EQUATION

Now, we consider the Poisson equation (26) in a slightly more abstract manner: Showing that a given Fisher-Rao flow $\rho_t$ is also a Wasserstein absolutely continuous curve requires solving the following (family of) PDEs

$$-\nabla \cdot (\rho_t \nabla s_t) = (\alpha_t - \overline{\alpha_t}) \, \rho_t, \quad t \in [0, 1], \tag{29}$$

for the vector field $v_t = \nabla s_t$ satisfying a suitable integrability condition. Here, $\alpha \in L_2(\boldsymbol{\rho})$, $\alpha_t := \alpha(t, \cdot)$ and $\overline{\alpha_t} := \mathbb{E}_{\rho_t}[\alpha_t]$ are given.

In order to do that, we look to find a weak solution to (29) in the space $\mathcal{X}_0^1$.

**Definition A.12.** *An element $\phi \in \mathcal{X}_0^1$ is called a weak solution to* (29) *if it satisfies,*

$$\int_0^1 \int_{\mathbb{R}^d} \langle \nabla \phi_t, \nabla \psi_t \rangle \, d\rho_t dt = \int_0^1 \int_{\mathbb{R}^d} \psi_t (\alpha_t - \overline{\alpha_t}) \, d\rho_t dt, \quad \forall \psi \in \mathcal{X}_0^1. \tag{30}$$

*Note that for $\psi \in C_c^\infty((0, 1) \times \mathbb{R}^d)$ it holds for $g(t) := \mathbb{E}_{\rho_t}[\psi(t, \cdot)]$ that $\psi - g \in \mathcal{X}_0^1$. Since $\nabla_x(g) = 0 = \int_0^1 \int_{\mathbb{R}^d} g(t)(\alpha_t - \overline{\alpha_t}) \, d\rho_t dt$, we have that* (30) *is also fulfilled for all $\psi \in C_c^\infty((0, 1) \times \mathbb{R}^d)$.*

We can now prove Theorem 1 from Section 2.2, which provides a solution to the Poisson problem (29) globally in time $t \in [0, 1]$.

**Theorem A.13.** *Assume that $\boldsymbol{\rho}$ from Definition A.1 satisfies* (PPI) *for some $K > 0$, and that $\frac{1}{\boldsymbol{\rho}} \in L_{1,\mathrm{loc}}([0, 1] \times \mathbb{R}^d)$. Furthermore, assume that $\alpha_t - \overline{\alpha_t} \in L_2(\boldsymbol{\rho})$, i.e., $\|\alpha_t - \overline{\alpha_t}\|_{L_2(\boldsymbol{\rho})}^2 \leq C < \infty$. Then, there exists a unique weak solution $s \in \mathcal{X}_0^1$ to* (29). *Moreover, $s$ satisfies the following inequality*

$$\int_0^1 \int_{\mathbb{R}^d} \|\nabla s\|^2 \, d\rho_t dt \leq K \int_0^1 \int_{\mathbb{R}^d} (\alpha_t - \overline{\alpha_t})^2 \, d\rho_t dt. \tag{31}$$

*Proof.* Let $A : \mathcal{X}_0^1 \to \mathbb{R}$ be defined as

$$A\psi := \int_0^1 \int_{\mathbb{R}^d} \psi_t (\alpha_t - \overline{\alpha_t}) \, d\rho_t dt.$$

Therefore, we can rewrite our system of PDEs (29) as

$$\langle s, \psi \rangle_{\mathcal{X}_0^1} = A\psi, \quad \forall \psi \in \mathcal{X}_0^1. \tag{32}$$

Clearly, $A$ is a linear operator. Since $\boldsymbol{\rho}$ satisfies (PPI) the following holds:

$$|A\psi|^2 = \left| \int_0^1 \int_{\mathbb{R}^d} \psi_t \left( \alpha_t - \overline{\alpha_t} \right) \mathrm{d}\rho_t \mathrm{d}t \right|^2 = \left| \int_{[0,1] \times \mathbb{R}^d} \psi_t(x) \left( \alpha_t(x) - \overline{\alpha_t}(x) \right) \mathrm{d}\boldsymbol{\rho}(t, x) \right|^2$$

$$\leq \left( \int (\alpha_t - \overline{\alpha_t})^2 \, \mathrm{d}\boldsymbol{\rho} \right) \cdot \left( \int |\psi_t|^2 \, \mathrm{d}\boldsymbol{\rho} \right) \leq C \int_0^1 \int_{\mathbb{R}^d} |\psi_t|^2 \, \mathrm{d}\rho_t \mathrm{d}t$$

$$\leq C \, K \int_0^1 \int_{\mathbb{R}^d} \|\nabla \psi_t\|^2 \, \mathrm{d}\rho_t \mathrm{d}t = C \, K \|\psi\|_{\mathcal{X}_0^1}^2.$$

By Lemma A.8, the space $\mathcal{X}_0^1$ is a Hilbert space. Applying the Riesz Representation theorem, there exists a unique element $s \in \mathcal{X}_0^1$ satisfying (32). $\qquad \square$

**Remark A.14.** *Note that in the case of Fisher-Rao flows, $\alpha$ satisfies*

$$\partial_t \rho_t = (\alpha_t - \overline{\alpha_t}) \, \rho_t, \quad t \in [0, 1].$$

*Then, the* Fisher-Rao action $\mathcal{A}_{\mathrm{FR}}(\rho_t)$ *of $\rho_t$ is given by* $\int_0^1 \|\alpha_t - \overline{\alpha_t}\|_{L_2(\rho_t)}^2 \, \mathrm{d}t$. *Hence, the r.h.s. of* (31) *can be described by* $K \, \mathcal{A}_{\mathrm{FR}}(\rho_t)$.

**Lemma A.15.** *Let $\boldsymbol{\rho}$ be such that $\rho_t$ is bounded (from above) for a.e. $t \in [0, 1]$. Then, for $s \in \mathcal{X}_0^1$ we have that $\nabla s_t \in T_{\rho_t} \mathcal{P}_2(\mathbb{R}^d)$ for a.e. $t \in [0, 1]$.*

*Proof.* For a.e. $t$, since $\rho_t$ is bounded from above by assumption, $C_c^\infty(\mathbb{R}^d)$ is dense in $H^1(\rho_t)$. Now, $s_t \in H^1(\rho_t)$ immediately yields the claim $\nabla s_t \in T_{\rho_t} \mathcal{P}_2(\mathbb{R}^d) = \overline{\{\nabla \phi : \phi \in C_c^\infty(\mathbb{R}^d)\}}^{L_2(\rho_t)}$. $\qquad \square$

It follows the proof of Theorem 2 from Section 2.2, which says that certain Fisher-Rao curves are indeed also Wasserstein absolutely continuous curves.

**Theorem A.16.** *Let $\rho_0, \rho_1$ be probability densities on $\mathbb{R}^d$ and assume that $\rho_t$ is a Fisher-Rao curve defined by*

$$\partial_t \rho_t = (\alpha_t - \overline{\alpha_t}) \rho_t,$$

*such that $\boldsymbol{\rho}$ and $\alpha$ satisfy the assumptions of Theorem A.13.*

*Then, $\rho_t$ is a Wasserstein absolutely continuous curve with vector field $\nabla_x s$, where $s \in \mathcal{X}_0^1$ is the unique weak solution of (30). If $\rho_t$ is bounded for a.e. $t$, we have that $\nabla_x s_t \in T_{\rho_t} \mathcal{P}_2(\mathbb{R}^d)$ for a.e. $t \in [0, 1]$, i.e., for every other vector field $v_t$ satisfying the continuity equation for $\rho_t$, we have that $\|v_t\|_{L_2(\rho_t)} \geq \|\nabla_x s_t\|_{L_2(\rho_t)}$ for a.e. $t \in [0, 1]$.*

*Furthermore, it holds that*

$$\int_0^1 \int_{\mathbb{R}^d} \|\nabla_x s_t\|^2 \, \mathrm{d}\rho_t \mathrm{d}t \leq K \int_0^1 \int_{\mathbb{R}^d} (\alpha_t - \overline{\alpha_t})^2 \, \mathrm{d}\rho_t \mathrm{d}t. \tag{33}$$

Note that by the *Benamou-Brenier formula*, see (Ambrosio et al., 2005, Equation 8.0.3), the left-hand side of (33) is an upper bound for the Wasserstein distance $W_2^2(\rho_0, \rho_1)$, while the double-integral on the right-hand side defines the Fisher-Rao action $\mathcal{A}_{\mathrm{FR}}$ of $\rho_t$.

*Proof.* Using $(\alpha_t - \overline{\alpha_t}) \rho_t = \partial_t \rho_t$ and Theorem A.13, we can conclude that $\rho_t$ satisfies the continuity equation

$$\partial_t \rho_t + \nabla \cdot (\rho_t v_t) = 0 \tag{34}$$

with a Borel vector field $v_t := \nabla s_t$ in the sense of distributions. Note that by construction, $s \in \mathcal{X}_0^1$, and hence, the gradient $\nabla_x s$ is already a Borel-measurable vector field on $[0, 1] \times \mathbb{R}^d$ such that $\nabla_x s_t \in L_2(\rho_t)$ for a.e. $t \in [0, 1]$. Further, (31) implies that $t \mapsto \|\nabla s_t\|_{L_2(\rho_t)} \in L^1([0, T])$.

Together with the fact that $\rho_t$ is weakly continuous, we infer by (Ambrosio et al., 2005, Theorem 8.3.1) that $\rho_t$ is Wasserstein absolutely continuous. By Lemma A.15, it holds that $\nabla s_t \in T_{\rho_t} \mathcal{P}_2(\mathbb{R}^d)$, if $\rho_t$ is bounded for a.e. $t$. In this case, (Ambrosio et al., 2005, Proposition 8.4.5) immediately implies that $\nabla s_t$ has minimal norm $\|\nabla s_t\|_{L_2(\rho_t)}$ among all Borel vector fields $v_t$ fulfilling (34). $\qquad \square$

Let us apply the above result to the curve given by linear interpolation of the energy functions $f_0, f_1$.

**Corollary A.17.** *Let $f_0, f_1 \in C^2(\mathbb{R}^d)$ be lower bounded by some $\gamma \in \mathbb{R}$ such that the conditions* (27) *and* (28) *of Propostion A.10 are satisfied for some $\alpha, \eta > 0$. Further, for simplicity, assume that $f_0, f_1$ have at most polynomial growth outside a ball, i.e., there exists $R > 0$ such that*

$$|f_i(x)| \le M\|x\|^k \quad \text{for all } \|x\| > R, \; i = 0, 1. \tag{35}$$

*Then, Theorem A.16 can be applied to the Fisher-Rao curve $\rho_t$ given by $V_t = (1-t)f_0 + tf_1$ and $\rho_t = Z_t^{-1} e^{-V_t}$ with $Z_t = \int_{\mathbb{R}^d} e^{-V_t} \, \mathrm{d}x$.*

*Proof.* First, by the drift condition (27) and linear growth (28), the measure $\boldsymbol{\rho}$ satisfies (PPI) by Proposition A.10. Furthermore, in Proposition A.10 we showed that $t \mapsto Z_t^{-1}$ is continuous on $[0, 1]$, hence $(t, x) \mapsto \rho_t(x) = Z_t^{-1} e^{-V_t(x)}$ is strictly positive and continuous on $[0, 1] \times \mathbb{R}^d$. Therefore, it immediately follows that $\frac{1}{\boldsymbol{\rho}} \in L_{1,\mathrm{loc}}([0, 1] \times \mathbb{R}^d)$.

We are left to check the $L_2$-condition on $\alpha_t = -\partial_t V_t = f_0 - f_1$. First, since $t \mapsto Z_t^{-1}$ is continuous on $[0, 1]$, hence bounded, it follows together with (28) and (35) that

$$
\begin{aligned}
|\overline{\alpha_t}| &\le Z_t^{-1} \int_{\mathbb{R}^d} |f_0 - f_1| e^{-V_t} \, \mathrm{d}x \\
&\le Z_t^{-1} \int_{\|x\| \le R} |f_0 - f_1| e^{-V_t} \, \mathrm{d}x \; + Z_t^{-1} \int_{\|x\| > R} |f_0 - f_1| e^{-V_t} \, \mathrm{d}x \\
&\le C_1 \int_{\|x\| \le R} |f_0 - f_1| e^{-\gamma} \, \mathrm{d}x \; + C_2 M \int_{\|x\| > R} \|x\|^k e^{-\alpha\|x\|} \, \mathrm{d}x \\
&\le C < \infty,
\end{aligned}
$$

with a constant $C > 0$ not depending on $t$. Hence, $\overline{\alpha_t} \in L_2(\boldsymbol{\rho})$, and we are left to prove $\alpha_t \in L_2(\boldsymbol{\rho})$. But this follows by the similar arguments

$$
Z_t^{-1} \int_{\mathbb{R}^d} |f_0 - f_1|^2 e^{-V_t} \, \mathrm{d}x \le Z_t^{-1} \int_{\|x\| \le R} |f_0 - f_1|^2 e^{-V_t} \, \mathrm{d}x \; + Z_t^{-1} \int_{\|x\| > R} |f_0 - f_1|^2 e^{-V_t} \, \mathrm{d}x
$$

$$
\begin{aligned}
&\le C_1 \int_{\|x\| \le R} |f_0 - f_1|^2 e^{-\gamma} \, \mathrm{d}x \; + C_2 M^2 \int_{\|x\| > R} \|x\|^{2k} e^{-\alpha\|x\|} \, \mathrm{d}x \\
&\le C < \infty,
\end{aligned}
$$

again, with a constant $C > 0$ not depending $t$. Integrating over $t \in [0, 1]$ yields $\alpha_t \in L_2(\boldsymbol{\rho})$, and hence, all assumptions of Theorem A.16 are fulfilled. □

**Remark A.18.** *Hence, the Fisher-Rao curves defined by linear interpolation $V_t = (1-t)f_0 + tf_1$ are absolutely continuous in the Wasserstein sense, e.g., in the cases (see also Remark A.11)*

- *Gaussian distributions given by $f_i := \frac{1}{2\sigma^2}\|x - m\|^2$,*

- *Laplacian distributions given by $f_i := \frac{1}{\sigma}\|x - m\|$,*

- *Laplacian type distributions given by $f_i := \min\{\|x\|, \|x - m\|\}$.*

Let us briefly discuss the differences between the theory developed here, and the prior results from Laugesen et al. (2014); Vargas et al. (2024).

- In Vargas et al. (2024), they assume that $\rho_t \in C^\infty([0, 1], \mathbb{R}^d)$ and $\alpha \in C^\infty([0, 1] \times \mathbb{R}^d)$ and a Poincaré inequality for each time $t$. We need much weaker conditions: we assume that $t \mapsto \rho_t$ is weakly continuous, $\rho = \int \rho_t dt$ fulfills a (partial in $x$) Poincaré inequality and $\alpha \in L^2(\rho)$. In particular, we allow single time points $t$, where the Poincaré inequality might fail for $\rho_t$.
- Our solution $\phi$ is an element of $H^1(\boldsymbol{\rho}) \subset L^2([0, 1] \times \mathbb{R}^d; \boldsymbol{\rho})$ and thus automatically measurable on the whole time domain $t \in [0, 1]$. This makes it possible to show that $\nabla \phi_t(x)$

is a Borel vector field with regularity in time which implies absolute continuity of the flow $\rho_t$ in the Wasserstein geometry. In contrast, Vargas et al. (2024) only considers a solution for every single $t \in [0, 1]$ without any regularity in $t$. This is a result already shown in (Laugesen et al., 2014, Thm. 2.2) and motivated our investigation into a framework across the time $t$.

- Furthermore, we give precise mild assumptions under which the partial Poincaré inequality is satisfied. In particular, we cover the case of the linear interpolation. As a result we obtain that these curves are absolutely continuous in the Wasserstein geometry.

## B  TELEPORTATION ISSUE WITH THE LINEAR INTERPOLATION

In this section we want to demonstrate that the norm $\|v_t\|_{L_2(\rho_t)}$ of the (optimal) velocity field (4) belonging to the Fisher-Rao path given by linear interpolation $f_t := (1 - t)f_0 + tf_1$, explodes for times $t$ close to 1, when the measures $\rho_0$ and $\rho_1$ are asymmetrical to each other.

In the following, we restrict ourselves to the one-dimensional setting $\mathbb{R}^1$, in order to exploit the isometry from the Wasserstein space $\mathcal{P}_2(\mathbb{R}^1)$ into the convex cone $\mathcal{C} \subset L_2(0, 1)$ of *quantile functions*, also see (Duong et al., 2024, Section 3). For the sake of self-containedness, we give a short introduction of the necessary tools: For $\mu \in \mathcal{P}_2(\mathbb{R})$ we define its cumulative distribution function (CDF) by

$$F_\mu(x) := \mu(-\infty, x], \quad x \in \mathbb{R},$$

and its left-continuous quantile function

$$Q_\mu(s) := \min\{x \in \mathbb{R} : F(x) \geq s\}, \quad s \in (0, 1).$$

Then, it holds the isometric property

$$W_2^2(\mu, \nu) = \int_0^1 |Q_\mu(s) - Q_\nu(s)|^2 \, \mathrm{d}s \quad \text{for all } \mu, \nu \in \mathcal{P}_2(\mathbb{R}). \tag{36}$$

Now, by (Ambrosio et al., 2005, Theorem 8.3.1), the norm $\|v_t\|_{L_2(\rho_t)}$ of the optimal field $v_t$ belonging to an absolutely continuous Wasserstein curve $\rho_t$ can be characterized by the *metric derivative* of the curve via

$$\|v_t\|_{L_2(\rho_t)} = |\rho'(t)| := \lim_{s \to t} \frac{W_2(\rho_t, \rho_s)}{|t - s|} \quad \text{for a.e. } t \in [0, 1].$$

Using the aforementioned isometry (36) and Komura's theorem, it holds for a.e. $t \in [0, 1]$

$$\lim_{s \to t} \frac{W_2(\rho_t, \rho_s)}{|t - s|} = \lim_{s \to t} \frac{\|Q_{\rho_t} - Q_{\rho_s}\|_{L_2(0,1)}}{|t - s|} = \|\partial_t(Q_{\rho_t})\|_{L_2(0,1)}.$$

Hence, if we can explicitly compute the quantiles $Q_{\rho_t}$ (and its time derivative), we get an analytic representation of $\|v_t\|_{L_2(\rho_t)}$ via

$$\|v_t\|_{L_2(\rho_t)}^2 = \int_0^1 \left(\partial_t(Q_{\rho_t})(s)\right)^2 \mathrm{d}s. \tag{37}$$

In order to do so, we choose the prior and target measures the following way: Let $f_0(x) := |x|$ and $f_1(x) := 2\min\{|x|, |x - m|\}$, where the second mode $m > 0$ is fixed for the moment. By Corollary A.17 and Remark A.18, the Fisher-Rao curve $\rho_t$ given by linear interpolation of $f_0, f_1$ is Wasserstein absolutely continuous. Now, it is straightforward (but tedious) to explicitly calculate the CDF and quantile function of $\rho_t$:

First, since $\rho_t \propto \rho_0^{1-t}\rho_1^t$ and $2\min\{|x|, |x - m|\} = |x| + |x - m| - \big||x| - |x - m|\big|$, we obtain

$$\rho_t \propto e^{-|x| - t|x - m| + t\big||x| - |x - m|\big|} \quad \text{for all } x \in \mathbb{R}, \ t \in [0, 1],$$

up to a normalization constant $Z_t$ which we calculate below.

**Calculating the CDF of $\rho_t$.** By dissolving the absolute values on suitable subintervals, we obtain for the CDF

$$Z_t \cdot F_{\rho_t}(x) = \frac{1}{1+t}e^{x(1+t)}, \quad x \le 0,$$

$$Z_t \cdot F_{\rho_t}(x) = \frac{2}{1+t} - \frac{1}{1+t}e^{-x(1+t)}, \quad 0 \le x \le \frac{m}{2},$$

$$Z_t \cdot F_{\rho_t}(x) = \frac{2}{1+t} - \left(\frac{1}{1+t} + \frac{1}{-1+3t}\right)e^{-\frac{m}{2}(1+t)} + \frac{1}{-1+3t}e^{x(-1+3t)-2tm}, \quad \frac{m}{2} \le x \le m,$$

$$Z_t \cdot F_{\rho_t}(x) = \frac{2}{1+t} - \left(\frac{1}{1+t} + \frac{1}{-1+3t}\right)e^{-\frac{m}{2}(1+t)}$$
$$+ \left(\frac{1}{1+t} + \frac{1}{-1+3t}\right)e^{-m(1-t)} - \frac{1}{1+t}e^{-x(1+t)+2tm}, \quad x \ge m,$$

where the normalization constant is hence given by

$$Z_t = Z_t \cdot F_{\rho_t}(\infty) = \frac{2}{1+t} - \left(\frac{1}{1+t} + \frac{1}{-1+3t}\right)e^{-\frac{m}{2}(1+t)}$$
$$+ \left(\frac{1}{1+t} + \frac{1}{-1+3t}\right)e^{-m(1-t)}.$$

**Calculating the quantile function of $\rho_t$.** Since $F_{\rho_t}(\cdot)$ is continuous and strictly increasing, its inverse is given by $F_{\rho_t}^{-1} = Q_{\rho_t}$. A simple calculation now yields

$$Q_{\rho_t}(s) = \frac{1}{1+t}\ln\left(Z_t(1+t)s\right), \quad 0 < s \le s_1,$$

$$Q_{\rho_t}(s) = -\frac{1}{1+t}\ln\left(-Z_t(1+t)s + 2\right), \quad s_1 \le s \le s_2,$$

$$Q_{\rho_t}(s) = \frac{\ln\left((-1+3t)(sZ_t - \frac{2}{1+t} + (\frac{1}{1+t} + \frac{1}{-1+3t})e^{-\frac{m}{2}(1+t)})\right) + 2tm}{-1+3t}, \quad s_2 \le s \le s_3,$$

$$Q_{\rho_t}(s) = \frac{-\ln\left(Z_t(1+t)(1-s)\right) + 2tm}{1+t}, \quad s_3 \le s < 1.$$

Here, the subintervals are given by

$$s_1 = \frac{1}{Z_t(1+t)},$$

$$s_2 = \frac{1}{Z_t}\left(\frac{2}{1+t} - \frac{1}{1+t}e^{-\frac{m}{2}(1+t)}\right),$$

$$s_3 = \frac{1}{Z_t}\left(\frac{2}{1+t} - \left(\frac{1}{1+t} + \frac{1}{-1+3t}\right)e^{-\frac{m}{2}(1+t)} + \frac{1}{-1+3t}e^{-m(1-t)}\right).$$

**Calculating the time derivative $\partial_t Q_{\rho_t}$ and the norm $\|v_t\|_{L_2(\rho_t)}^2$.** Using standard differentiation rules, the time derivative of $Q_{\rho_t}$ can be computed separately on each subinterval $(s_i, s_{i+1})$. Since the resulting derivatives become unbearably long, we only include them in the Supplementary Material. Nevertheless, they can be computed analytically as elementary functions, and be inserted into the desired integral (37). By numerical integration, we can finally approximate the norm $\|v_t\|_{L_2(\rho_t)}^2$ arbitrarily close. The computed results are depicted in Figure 2 for different values of the mode $m > 0$; roughly speaking, for late times $t \sim 1$, the norm of the velocity field grows *exponentially* with $m$.

On a side note, the integration of $\partial_t Q_{\rho_t}$ *only* on the subintervals $(s_1, s_2)$ and $(s_2, s_3)$ contributes to the explosion of $\|v_t\|_{L_2(\rho_t)}^2$. Intuitively, this corresponds to the fact that mass gets shifted very abruptly *mainly* in the areas $x \in (0, \frac{m}{2})$ and $(\frac{m}{2}, m)$ for late times $t$.

**On related work.** The teleportation issue described here has been discussed in other works under the term *mode switching* or *label switching*, see, e.g., Woodard et al. (2009); Syed et al. (2022); Phillips et al. (2024); Chehab & Korba (2024); Noble et al. (2024). We complement these mostly empirical observations by theoretical results.

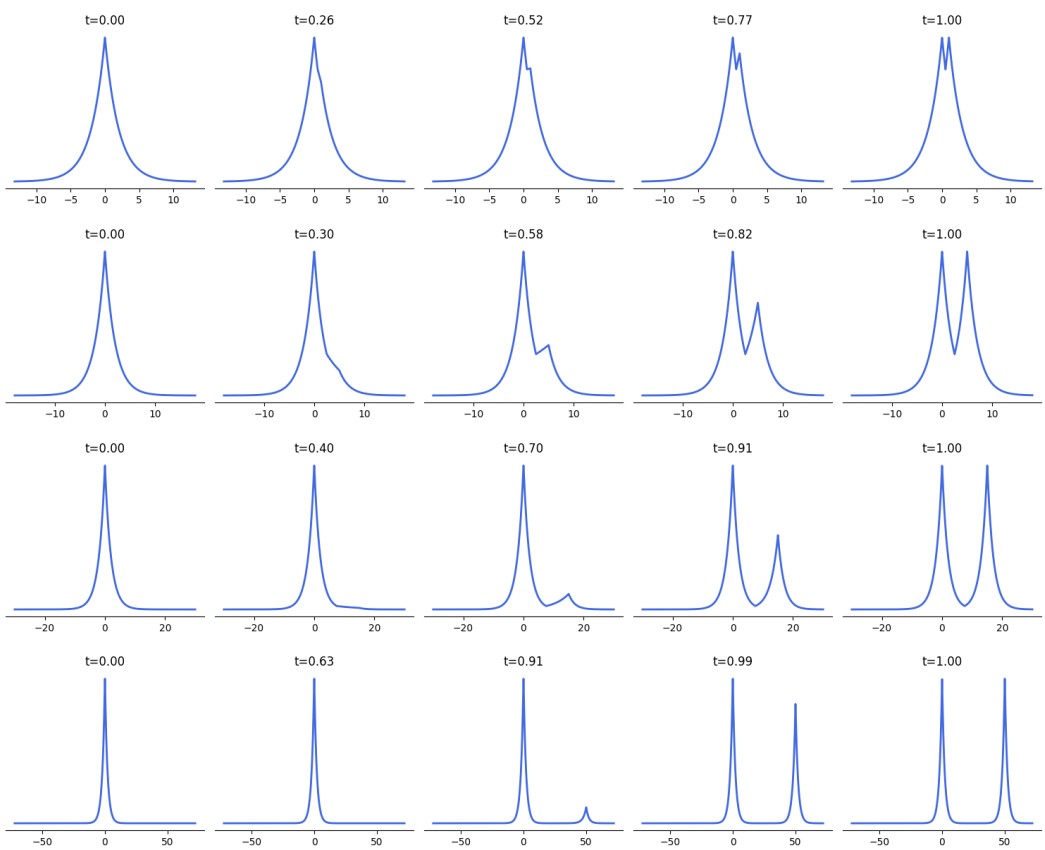

Figure 5: Evolution of probability densities $\rho_t$ for $m \in \{1, 5, 15, 50\}$.

## C  ALGORITHMS

---

**Algorithm 1** Learning $v_t^\theta, C_t^\theta$ in (17) for $x^i$ sampled from trajectories.

---

**Parameters and Networks:** $\theta = (\theta_1, \theta_2)$ and $v_t^{\theta_1}, C_t^{\theta_2}$.
**Functions :** $f_t = (1 - t)f_0 + tf_1$, $v_t^{\theta_1}$, $C_t^{\theta_2}$.
**Loss:** $\mathcal{E}(\theta, x, t) := |f_1 - f_0 - C_t^{\theta_2} + \langle \nabla f_t, v_t^{\theta_1} \rangle - \nabla \cdot v_t^{\theta_1}|^2$
**Algorithm:**
**for** $i = 1, ..., N$ **do**
 grad = 0
 **for** $l = 1, ..., n$ **do**
  Draw $x^j \sim \rho_Z, 1 \le j \le m$
  ODEsolve $x_{t_l}^j$ as solution of $\partial x_t^j = v_t^\theta(x_t^j), x_0^j = x^j$ at time points $t_l \in [0, 1]$
  Detach $x_{t_l}^j$ from computational graph
  grad+= $\nabla_\theta \sum_{i=1}^n \sum_{j=1}^m \mathcal{E}(\theta, x_{t_l}^j, t_l)$
 **end for**
 Update $v_t^{\theta_1}, C_t^{\theta_2}$ using grad
**end for**
**Output:** $v_t^{\theta_1}, C_t^{\theta_2}$.

---

**Algorithm 2** Learning $f_t^{\theta_1}, v_t^{\theta_2}, C_t^{\theta_3}$ in (18) for $x^i$ sampled from trajectories.

---

**Parameters and Networks:** $\theta = (\theta_1, \theta_2, \theta_3)$ and $\psi_t^{\theta_1}, v_t^{\theta_2}, C_t^{\theta_3}$.
**Functions:** $f_t^{\theta_1} = (1 - t)f_0 + tf_1 + (1 - t)t\psi_t^{\theta_1}$, $v_t^{\theta_2}$, $C_t^{\theta_3}$.
**Loss:** $\mathcal{E}(\theta, x, t) := |\partial_t f_t^{\theta_1}(x) - C_t^{\theta_3} + \langle \nabla f_t^{\theta_1}, v_t^{\theta_2} \rangle - \nabla \cdot v_t^{\theta_2}|^2$
**Algorithm:**
**for** $i = 1, ..., N$ **do**
 grad = 0
 **for** $l = 1, ..., n$ **do**
  Draw $x^j \sim \rho_Z, 1 \le j \le m$
  ODEsolve $x_{t_l}^j$ as solution of $\partial x_t^j = v_t^\theta(x_t^j), x_0^j = x^j$ at time points $t_l \in [0, 1]$
  Detach $x_{t_l}^j$ from computational graph
  grad+= $\nabla_\theta \sum_{i=1}^n \sum_{j=1}^m \mathcal{E}(\theta, x_{t_l}^j, t_l)$
 **end for**
 Update $f_t^{\theta_1}, v_t^{\theta_2}, C_t^{\theta_3}$ using grad
**end for**
**Output:** $f_t^{\theta_1}, v_t^{\theta_2}, C_t^{\theta_3}$.

---

**Algorithm 3** Learning $\psi_t^{\theta_1}, C_t^{\theta_2}$ in (19) as done in Section 4.

---

**Networks:** $\psi_t^{\theta_1}, C_t^{\theta_2}$
**Parameters and functions:** $\theta = (\theta_1, \theta_2)$ and $f_t = (1 - t)f_1 + t\psi_t^{\theta_1}$, $C_t^{\theta_2}$.
**Loss:** $\mathcal{E}(\theta, x, t) := |\partial_t f_t^{\theta_1} - C_t^{\theta_2} + \langle \nabla f_t^{\theta_1}, \nabla(f_t^{\theta_1} - f_Z) \rangle - \Delta(f_t^{\theta_1} - f_Z)|^2$
**Algorithm:**
**for** $i = 1, ..., N$ **do**
 Draw $x_1^j \sim \rho_Z, 1 \le j \le m$ and $x_0^j \in \mathcal{U}_{a,b}, 1 \le j \le m$ uniformly
 Draw $t_j \in [0, 1], 1 \le j \le m$ uniformly and use (25) to set $x_{t_j}^j$
 grad = $\nabla_\theta \sum_{j=1}^m \mathcal{E}(\theta, x_{t_j}^j, t_j)$
 Update $\psi_t^{\theta_1}, C_t^{\theta_2}$ using grad
**end for**
**Output:** $\psi_t^{\theta_1}, C_t^{\theta_2}$.

---

## D ADDITIONAL DETAILS ON THE EXPERIMENTS

**Implementation and evaluation details.** We implement the algorithms in Pytorch Paszke et al. (2019). For both experiments we use standard MLPs with "swish" activation functions. For the linear and learned interpolations we parameterize the vector fields directly. The learned interpolation also uses the same network for $\psi$. The linear interpolation uses the same network for the time dependent function $C_t$. For the learned and gradient flow interpolations we implement $C_t$ using the same network with less parameters. The gradient flow interpolation also uses a smaller network for the time scheduling function $g$ as defined in Section 4. In case of normalized energy functions we can omit the parametrization of $C_t$ for the learned and gradient flow interpolations. We simulate the corresponding ODEs using the torchdiffeq Chen (2018) package with the Runge-Kutta adaptive step size solver ("dopri5"). For the linear and learned interpolation we use $50$ time steps along which the loss is computed and the gradients are accumulated with a batch size of $256$. For the gradient flow interpolation we sample $4096$ particles at random uniform time points and therefore do not accumulate gradients. We adjusted the number of iterations such that all methods ran approximately the same time on the same hardware.

We compute log weights of the generated samples by using the continuous change of variables formula and the corresponding implementation from Chen (2018). We evaluate the methods by generating $5 * 10^4$ samples and log weight and computing the effective sample size, negative log likelihood and energy distance. We report mean and standard deviation over $10$ evaluation runs. The effective sample size is computed as in Midgley et al. (2023). We compute the energy distance using the GeomLoss library Feydy et al. (2019). For the target energy functions and plotting utilities we use the code provided in Midgley et al. (2023).

The gradient flow interpolation faced some difficulty in the many well experiment when, while sampling, particles entered regions of very low density. This could severely affect sampling times when using the adaptive step size solver. Therefore we also used an Euler scheme and resampled a given batch if a NaN error is raised. This happens approximately for $0.01\%$ of the particles.

**Choice of latent distribution.** The linear and learned interpolations depend significantly on the choice of initial distribution. In the following we visualize and compare numerically the results when starting in Gaussians with different standard deviations. Note that when using a small standard deviation not all modes are recovered. On the other hand choosing a very large standard deviation leads to the modes close to $0$ not receiving enough mass.

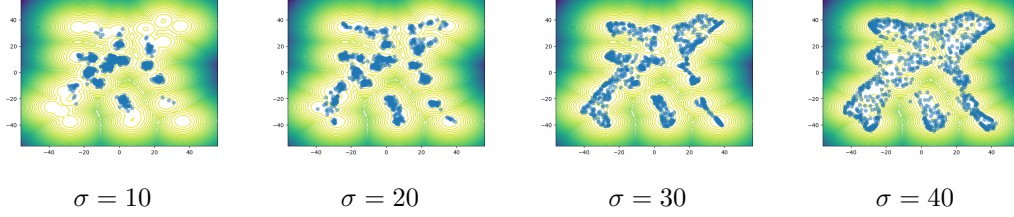

| $\sigma = 10$ | $\sigma = 20$ | $\sigma = 30$ | $\sigma = 40$ |

Figure 6: Results for the linear interpolation for different values of $\sigma$.

| | ESS $\uparrow$ | NLL $\downarrow$ | Energy $\downarrow$ |
|---|---|---|---|
| $\sigma = 10$ | $0.002 \pm 0.003$ | $7.071 \pm 0.005$ | $2.887 \pm 0.021$ |
| $\sigma = 20$ | $0.063 \pm 0.07$ | $7.362 \pm 0.005$ | $0.592 \pm 0.005$ |
| $\sigma = 30$ | $0.183 \pm 0.05$ | $8.657 \pm 0.012$ | $0.247 \pm 0.016$ |
| $\sigma = 40$ | $0.092 \pm 0.001$ | $12.909 \pm 0.023$ | $0.554 \pm 0.008$ |

Table 3: Comparison of effective sample size, negative log likelihood, and the energy distance for the linear interpolation with initial distribution being a Gaussian with standard deviation $\sigma$.

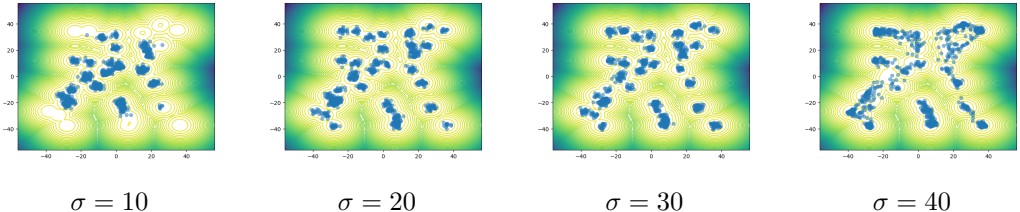

$$\sigma = 10 \qquad\qquad \sigma = 20 \qquad\qquad \sigma = 30 \qquad\qquad \sigma = 40$$

Figure 7: Results for the learned interpolation for different values of $\sigma$.

|  | ESS ↑ | NLL ↓ | Energy ↓ |
|---|---|---|---|
| $\sigma = 10$ | $0.009 \pm 0.011$ | $7.275 \pm 0.005$ | $1.253 \pm 0.017$ |
| $\sigma = 20$ | $0.605 \pm 0.316$ | $6.894 \pm 0.005$ | $0.058 \pm 0.006$ |
| $\sigma = 30$ | $0.922 \pm 0.002$ | $6.913 \pm 0.003$ | $0.121 \pm 0.009$ |
| $\sigma = 40$ | $0.139 \pm 0.052$ | $7.779 \pm 0.012$ | $0.531 \pm 0.014$ |

Table 4: Comparison of effective sample size, negative log likelihood, and the energy distance for the learned interpolation with initial distribution being a Gaussian with standard deviation $\sigma$.

