# OpenReview forum: "Neural Sampling from Boltzmann Densities: Fisher-Rao Curves in the Wasserstein Geometry"
_ICLR.cc/2025/Conference — ICLR 2025 Poster_

### Official Review · Reviewer_CXu4 · 2024-10-28

**Soundness:** 3
**Presentation:** 2
**Contribution:** 3
**Rating:** 6
**Confidence:** 2

**Summary:**

The paper deals with the problem of sampling configuration from an un-normalized Boltzmann density. The author consider an approach where the energy function is interpolated by a function $f_t$ that is learned, following an approach developed by Máté and Fleuret. The main scope of the paper is to prove that the curve of the Boltzmann densities are absolutely continuous Wasserstein curve. They also demonstrate on an analytical example that the approach of Máté and Fleuret lead to a discontinuity in the particle transport due to the explosion of the velocity field. They propose a new reparametrization for the interpolant where a single function has to be learned while the velocity field is fixed.

**Strengths:**

The paper make very interesting connections between Wasserstein Flows and the developed method.
It also underlines its relation with Diffusion models.

**Weaknesses:**

The article is particularly technical for those not-familiar with Wasserstein flows etc. While I find the results very interesting, I'm confused about the overuse of mathematical technicality, which tends to make the paper hard to grasp for a non-specialist. I'm wondering if the authors could ease part of it (eventually keeping a lot of details in the appendix) but putting an emphasis on the general direction and method.

Among all these technicalities, I end up not understanding the details on how the function $\psi_t$ is learned, or the velocity in the "learned interpolation" case.

The results on the experiments are ok, but not super-convincing either.

**Questions:**

- The experiments that are done in Máte-Fleuret, show that problems of mode-collapse occur more severely when the Gaussian distributions are unevenly distributed. While the dataset proposed by the authors is not homogeneous since the mean of the Gaussians are random, it might be better to include a test on a mixture with unequal weights and variances.
- Does there exist a case where the gradient flow method is clearly better than the learned one ?
- There very few experimental validations, I would like to see another dataset, maybe inspired from the one investigated in Máté and Fleuret.

It seems to me that the phenomena described by the use of the linear interpolation, as in fig 1, is very similar to what is called "first order transition" in physics. As a parameter is slightly change, the overlap between the distribution $\rho_t$ and $\rho_{t+\delta t}$ is very small. That being said, the condition for annealed importance sampling to work is that the overlap between two nearest neighbors of the interpolation remain high enough. Is there a way to interpret the diverging of the velocity as characterize by the authors with such a phenomenon ?

Minor remark:
- there is a tipo at "Leibner" line 96

---

> ### Author Response · Authors · 2024-11-20
>
> Weakness:
> Sorry for the high mathematical technicality which are necessary here. We have already tried to put most of the theoretical analysis in the appendix.
> In order to make the learning part understandable without going into the mathematical details, we started adding pseudocode in Appendix C.
>
> We added the numerical examples.
>
> Questions:
>
>
> We added the numerical example you proposed in Example 5. Thanks!
> As expected, the linear interpolation fails, while the learned and the gradient flow interpolation properly approximate the target distribution.
>
> In addition we give a numerical example in 16 dimensions.
>
> For the 2d Gaussian mixture reported in our numerical section (Table 1) the gradient flow method performs better than the learned one.
>
>
>
> *First order transition*:
> There do seem to be some similarities to the physical phenomenon you described. In our phenomenon described in Figure 1, the distribution $\rho_t$ changes drastically to another distribution $\rho_{t + \delta}$ for late times $t \approx 1$, given that the overlap of the two distributions is very small, and the squared velocity $\|v_t\|_{L_2}^2$ explodes. This might resemble the discontinuity of the first derivative of the energy happening at first-order phase transitions.

---

> ### Comment · Reviewer_CXu4 · 2024-11-21
>
> I thank the author for his reply, and I would have raised my grade to 7 if that would be possible...

---

### Official Review · Reviewer_mKTY · 2024-10-29

**Soundness:** 3
**Presentation:** 2
**Contribution:** 3
**Rating:** 5
**Confidence:** 3

**Summary:**

The present paper considers the problem of sampling from a Boltzmann distribution known up to a normalization constant.

The author investigate theoretically the connection between the evolution of Boltzmann distributions with energy indexed by time, which give rise to Fisher Rao flow, and absolutely continuous curve in the Wasserstein space that obey a continuity equation. In particular, the author set the conditions under which a regular velocity field expressed as a gradient is solution to continuity equation associated with a the Boltzmann distribution curve.

Using this connection, the authors propose to learn a curve of Boltzmann densities bridging a base distribution to the target: assuming it follows a continuity equation for a velocity field expressed as a gradient a training objective is derived as in Mate & Fleuret (2023).  Going further, the authors take inspiration from the Fokker Plank equation of the Ornstein-Uhlenbeck process, recalling it’s interpretation as a Wassertein gradient flow on the reverse KL, to propose corresponding forms for the velocity field and learned energy. As such the learned curve is related to a variance preserving noising scheme of the target distribution.

Ultimately, the ODE associated with the velocity field can be integrated backward for sampling. The proposed method is compared against closely related methods that learn different interpolation schemes between a simple Gaussian base distribution and the target. The simple experiments in 2 and 8 dimensions suggest that the present method is less sensitive to the choice of the base distribution.

**Strengths:**

- The paper introduces well the concepts and challenges it aims to address, for example using the analytical example of Figure 1 and 2.
- The paper highlights and fruitfully exploits connections between SDE/ODE sampling and gradient flows on probability spaces.
- The paper draws its inspiration from prior literature, but draws new connections to propose amendments to the approach.
- The proposed sampling method appears to be superior to related ones as its initial distribution can be chosen as a standard Gaussian regardless of the variance/mode location in the target.

I am not familiar enough with the formal literature on Wasserstein curves to judge the novelty of the theoretical results of the paper.

**Weaknesses:**

- The writing of the paper could be improved, in particular, some notations are regularly used before being introduced.
- The discussion of related works should be extended to include:
	- Stochastic interpolants: https://arxiv.org/abs/2303.08797
	- Non-parametric denoising based samplers: RDMC https://openreview.net/forum?id=kIPEyMSdFV & SLIPS https://proceedings.mlr.press/v235/grenioux24a.html
- The experimental validation is currently very limited, notably in terms of how the method scales with increasing dimension.

Minor:
- maybe misses the assumption that $\rho_1 = \rho_D$?
- Notations in Equation (2) are not defined, also lacking Equation (5)
- In (17), the space variable is sampled uniformly on an hypercube? How are the boundaries chosen?


- typos:
	- line 96: Kullback Leibner
	- extra comma between $\rho_t$ and $v_t$ in the divergence (11)

**Questions:**

-

---

> ### Author Response · Authors · 2024-11-20
>
> Concerning the novelty of the
> theoretical results, please see our response to Reviewer m4s7 and their acknowledgement.
>
> Weakness 1:  We added certain notation to make it easier to read.
>
> Weakness 2: Thanks for suggesting the three references.  We included them in the discussion about related work. The first reference about stochastic interpolants investigates the setting, where samples from the source and target distribution are available, which is not the case in our setting. But it is interesting, since it is investigating different paths in probability space. In the RDMC paper, the score at point $x_t$ is estimated using a unadjusted Langevin algorithm. This score is then used to obtain $x_{t+h}$. They do not learn a vector field $v_t$, but have to rerun the entire algorithm if they want to draw new samples.  Similarly, in the paper SLIPS they use MCMC in order to approximate the drift of an SDE and use that approximation for sampling. This is an interesting approach, but again the goal here is not to learn a vector field that can be used for sampling.
>
> Weakness 3:
>     We added numerical results, so far up to dimension 16, and an example proposed by reviewer CXu4.
>
> Minor:
> We added the assumption $\rho_1 = \rho_d$.
>
> We added the notation in (2) and (5).
>
> We heuristically tried different domains, such that they are not too large which could lead to numerical instability and not too small to create a bias. We did not optimize over this choice.
>
> We added Remark 3 on sampling.
>
> We corrected the typos, thanks for pointing out!

---

> > ### Author Response · Authors · 2024-11-21
> >
> > We thank you again for your thorough review. We would like to also mention, with respect to the novelty of our theoretical results, that in our latest revision, we added a discussion of our theoretical contributions (at the end of Appendix A). In particular we compare these to the existing results of [laug,nusken].
> >
> > [laug]: R. S. Laugesen, P. G. Mehta, S. P. Meyn, and M. Raginsky. Poisson’s equation in nonlinear filtering.
> > arXiv preprint arXiv:1412.5845, 2014.
> >
> > [nusken] Nusken, N., Vargas, F., Padhy, S. and Blessing, D., 2024, May. Transport meets variational inference: Controlled monte carlo diffusions. In The Twelfth International Conference on Learning Representations: ICLR 2024.

---

> > ### Comment · Reviewer_mKTY · 2024-11-25
> >
> > Thank you for answering my questions and adding new experiments.
> >
> > Figure 3 suggests an advantage with respect to mode collapse, which is very interesting. Is this robustness to mode collapse robust to higher dimension ? The results from Table 2 does not allow to assess for this point. If I understand correctly, in dimension 16, the presented results suggest that the proposed method is not as efficient as the one with learned flow.

---

> > > ### Author Response · Authors · 2024-11-25
> > >
> > > Indeed, Figure 3 shows a crucial advantage of the gradient flow interpolation over the linear interpolation with respect to mode collapse. Since the gradient flow interpolation corresponds to a **spacial** interpolation, heuristically speaking the corresponding dynamics are "regular" in space. From a theoretical viewpoint this property should be valid in any dimension. Hence in higher dimension the gradient flow interpolation should *still* be more robust to mode collapse. Numerically, this is supported by the fact that the gradient flow path is applied in high dimensional applications such as image generation with high diversity using diffusion models. Note that for the 16 dimensional example the issue is not exactly mode collapse for any of the methods. It is more about a mismatch of the weights of the modes. As you have correctly stated the learned interpolation performs the best in this scenario. Still, there is a significant advantage of the gradient flow interpolation over the linear interpolation. But also note that the gradient flow interpolation is analytically accessible while the learned interpolation is a black box curve.

---

### Official Review · Reviewer_FHYM · 2024-11-04

**Soundness:** 4
**Presentation:** 3
**Contribution:** 3
**Rating:** 8
**Confidence:** 3

**Summary:**

The paper considers the problem of sampling from probability densities. The key idea is to use a so-called Fisher-Rao flow, which is a general equation that transports a simple density to a more complex one, going through a sequence of interpolating densities that are only assumed to be known up to an unknown normalizing constant. The main theoretical results of this paper (Theorem 1 and 2) show under very general conditions the uniqueness of a weak solution of the Poisson equation of this flow for which the Fisher-Rao curve is Wasserstein absolute continuous. The solution of this Poisson equation is found by parametrizing it using neural networks, and minimizing a certain loss function (explained on pages 5 and 6). Given this solution, the ODE of equation (6) is solved backward to yield samples from the target distribution by transporting a simple initial distribution to the target. Multiple different paths (interpolating densities) are considered, namely linear interpolation, and learned interpolation, where the interpolation curve itself is also parameterized by a neural network, and certain loss function enforcing more smooth transitions along the Fisher-Rao curve is optimized.

**Strengths:**

The theoretical results are quite fundamental, and show existence of unique solution to the Poisson equation that ensures Wasserstein absolute continuity. The idea of using a neural network to parameterize the interpolating distributions is also novel for Fisher-Rao flow, as far as I know. The numerical results sufficiently illustrate the methodology, and the choice of metrics makes sense.

**Weaknesses:**

There could be a more precise description of the ODE solver used to solve equation (6) backward, since this is the essence of the method for sampling from targets.
There is no theoretical guarantees for the neural network approximation that is used to solve (17) actually is able to solve it.
The scalability of this approach to high dimensional problems has not been demonstrated.

**Questions:**

Could you please explain in more detail the precise methodology used for solving the ODE (6)? Do you need a stiff ODE solver?

Both examples you've considered are 2 dimensional. Could you discuss the scalability of this methodology to higher dimensional problems?

---

> ### Author Response · Authors · 2024-11-20
>
> Weakness: We mainly use an adaptive Runge-Kutta-5 stepsize solver  for solving the ODE. It is true that solving this ODE is crucial and there are three main obstructions to overcome. First, we can empirically only learn the vector field on a bounded domain and thus outside of this domain the approximation might fail. If we then sample from a Gaussian and obtain a starting point outside this domain, the particle might be pushed in a wrong direction, which sometimes happens in practice.
>
> The second issue concerns the straightness of the ODE paths for the ground truth vector field, which affects how good the ODE solvers work. Both issues are interesting problems to analyze, but they are out of the scope of this paper. The third issue comes up when the vector field becomes irregular meaning, e.g. the $L^2$ norm of the vector field explodes as in the example ''A teleportation issue with the linear interpolation'' at p. 6. Such a vector field is then hard to approximate. We partly solve this problem by choosing the gradient flow interpolation that is in general better behaved as the linear interpolation of the energies.
>     We added numerical results, so far up to dimension 16, and an example proposed by reviewer CXu4.
>
> Question: Concerning the ODE solver, see our answer above.
> Please note that the Many Well experiment was already conducted in 8 dimensions in the original submission and we scaled it up to 16 dimensions now.

---

> > ### Comment · Reviewer_FHYM · 2024-11-23
> > **Response to Official Comment by Authors**
> >
> > After considering the authors response, my concerns about the scalability of this method have been lessened.
> > Clearly, there is still further research to be done on making this approach more robust and reliable and applicable to more complex datasets. Nevertheless, given that the paper has proven rather fundamental mathematical results related to diffusion models, I am raising my score to 8.

---

### Official Review · Reviewer_rTNR · 2024-11-07

**Soundness:** 2
**Presentation:** 2
**Contribution:** 2
**Rating:** 5
**Confidence:** 4

**Summary:**

The paper deals with the problem of sampling from unnormalized density (with known energy function). This is done by means of learning a specific Fisher-Rao curve in the probability measures space, which transforms the data distribution to some simple distribution (Gaussian one is considered). Since the vector field of this curve is learned, it could be inverted and simulated with ODE for sampling.

**Strengths:**

The paper has some interesting theoretical results about absolute continuity and optimality of Fisher-Rao flows. Also, the paper demonstrates the connection between Fisher-Rao flows and Wasserstein Gradient flows (w.r.t KL functional).

**Weaknesses:**

I think that the paper has rather limited methodological and practical contributions.
1. In fact, the authors take the method from [Mate&Fleuret] (in particular, eq. 15 is exactly eqs. 11 and 12 from [Mate&Fleuret]), but with specific parameterization of the vector field $v_t$. In turn, the chosen vector field parameterization exactly corresponds to Wasserstein gradient flows (WGF) with KL divergence. Therefore, the proposed method - modeling WGF with KL from data to Gaussian (with subsequent inversion) using  techniques from [Mate&Fleuret]. At this point I want to note that there are other papers which also solve something similar to WGF inversion task with different techniques (e.g., [Boffi], JKO: [Xu]). So, what is the advantage of your proposed method? In particular, [Xu] manages to work with image data, while I am not sure if the presented method scales to high dims.
2. The scalability of the method is under question. The experimental illustrations do not stress test high-dimensional applications.

**Questions:**

1. General background section. What do you denote by $C_{c}^{\infty}$? Is it the set of (infinitely-many) differentiable functions?
2. Lines 261 - 267. I missed whether you introduce your method in these lines, or just explain the method by [Mate&Fleuret]. But in the latter case, this loss is different from [Mate&Flauret], because instead of sampling $x \in U[a, b]^d$ uniformly, as in eq. (17), they propagate $x$ along the learned vector field (treated as continuous normalizing flow).
3. Lines 359-360. I do not understand, what you mean by “[...] it is not clear if Boltzmann densities stay Boltzmann densities”

**Misprints**

1. Line 187 - no comma in $\nabla \cdot (\rho_t, v_t)$ in eq. (11).
2. Line 037-038. Velocity field $v: [0, 1]\times \mathbb{R}^d \rightarrow \mathbb{R}$ maps to scalar? Misprint?

**Minor comments**

A bit difficult to read the text (Introduction section, Related works subsection). Line 118, eq. (2). What is $s_t$ in eq. (2), what is $\alpha_t$, $\overline{\alpha}_t$ in eq. (2)? All of these quantities are introduced later, in the main text, but I think it is strange to introduce a formula in the introduction which will be understandable only after reading the main text. Some references/brief explanations should be given.

**Related works**

[Mate&Fleuret] Mate et. al., Learning Interpolations between Boltzmann Densities, TMLR’2023

[Xu] Xu et. al., Normalizing flow neural networks by JKO scheme, NeurIPS’2023

[Boffi] Boffi et. al., Probability flow solution of the fokker--planck equation

---

> ### Author Response · Authors · 2024-11-20
>
> Thanks for acknowledging our nontrivial
> theoretical results.
> We think that having analytical evidence that the linear interpolation
> leads to exploding velocities (instead of just numerical observations) is important.
>
> Weakness 1: Thanks for pointing out the reference [Xu]. They also tackle the task of sampling from an unnormalized target density by approximating a Wasserstein gradient flow of the KL. However, their curve is the gradient  flow of $KL(\cdot|\rho_D)$, i.e. the flow from $\rho_Z$ to the data, while  we use the gradient flow of $KL(\cdot,\rho_Z)$ starting from $\rho_D$.  While they are using one network per gradient flow approximation step, we use one network for the whole flow.
>
>  The paper [Boffi]
>  considers a SDE $d x_t=b_t(x_t)dt + \nabla\cdot D_t(x_t)dt + \sqrt{2}\sigma_t dW_t$, where they assume that $b_t$ and $D_t$ are known. This in in contrast to our approach where we do not know $D_t$, but it rather depends on the curve $\rho_t$. Hence their approach is not applicable in our case. We added both reference in our related works section.
>
> Weakness 2: We added numerical results, so far up to dimension 16, and an example proposed by reviewer CXu4.
>
>
> *Questions*:
>
> - 1. $C_{c}^{\infty}$ denotes the set of infinitely-often differentiable functions with compact support. We added a corresponding definition in the updated version.
>     You are also right that in line 118, we should already introduce the notions of $s_t, \alpha_t$ etc. We will update this in our revised version.
>
> - 2. Yes, as mentioned the linear and the learned interpolation methods are known. Also, the approach leading to lines 261-267 for solving the PDE (16) was already proposed (see line 257) by [Mate\&Fleuret].
>     As you stated, [Mate\&Fleuret] sample $x$ from the trajectory of the learned vector field which we explained in lines 309-315. We  moved this comment further up right after Equation (17) to make the lines 261-267 clearer. In  Section 4, we experimented  using both, the uniform and trajectory based sampling strategy, for learning.  More details are given at the beginning of Section 4.
>
> - 3. Given a Boltzmann target distribution and the dynamics governed by the Fokker-Planck equation it is not a-priori clear that the intermediate densities $\rho_t$ themselves satisfy $\rho_t \propto e^{-f_t}$ for some $f_t$, in particular it is not clear that $\rho_t$ is strictly $>0$. This is however necessary, since $f_t$ is the quantity we aim to approximate. We  added further explanation to make this point clearer.
>
> Thanks, we corrected the two misprints!

---

> > ### Comment · Reviewer_rTNR · 2024-11-25
> > **Thanks to the authors**
> >
> > I thank the authors for their response, and for the undertaken fixes/modifications. I still think that the work, while theoretically interesting, still does not make may methodological and practical contributions. Based on this, I keep my score as it is.

---

### Official Review · Reviewer_m4s7 · 2024-11-11

**Soundness:** 3
**Presentation:** 4
**Contribution:** 2
**Rating:** 8
**Confidence:** 4

**Summary:**

This work contributions are twofold:

1. First, the work provides conditions under  which a probability flow admits a velocity field satisfying the continuity equation
2. Following from the theory this work then proposes learning the vector field via a PINN loss on the log of the continuity equation (PDE arising from taking the log of the probability flow) and explore several different interpolation schemes.

The work then presents some empirical results motivating the success of their proposed method.

**Strengths:**

The paper is rigorous and very well written and proposes a very sound approach for sampling from unnormalised densities based on the continuity equation, with strongly backed theoretical results and some modest numerical results; the paper also proposes a novel interpolation scheme (gradient flow interpolation) in the context of learning the vector field with promising numerical results.

In general I find the connections to Wasserstein gradient flows and the general formal machinery in the W_2 metric space to be a rather strong selling point of this work in contrast to prior works which explore learning the same vector field, during the review period I look forward to discussing/exploring this further with the authors, in order to improve the manuscript and highlight their contributions.

**Weaknesses:**

Unfortunately, this work misses prior work, which already explores learning the exact same vector field / continuity equation in the exact same context/task.

1.  Prior work [1]  (ICLR 2024 , ArXived July  2023)  provides a family of objectives for learning the vector field of the continuity equation in the exact same continuity equation + vector field up to a divergence. Notice [1] also provides an existence result for the vector field in the context of sampling from boltzman distributions (See Proposition 3.2 Appendix D3), the authors should make it clear how their results are different (from what I can see this submission has stronger / more general results, however they should be more precise in comparing to prior work). Note that whilst [1] has formulated their work in the SDE setting it is easy to see that their SDE in Equation 21 satisfies the same continuity equation as your work (See Equation 50 in their appendix) when using the optimal drift / velocity field considered in both works. Finally note that concurrent work [3] (Appendix 5.6 ) provides an explicit connection between [1] and your PINN based objective
2. Prior Work [2]  (workshop version [4] Published on March 2024 around 5 months before the ICLR deadline https://openreview.net/pdf?id=KwHPBIGkET) explores the exact same objective for learning (They call it ODE anneal, see [3] equation 26 for a more clear rewriting of the PINN objective in [2,4]). As you highlight [5] also explores this loss

So, unfortunately, I would say that works [1,2,4] cannot be deemed as concurrent and have already explored NN-based learning of the vector field in the continuity equation for sampling from Boltzmann densities.  Therefore, this work needs to both conceptually and empirically discuss/compare these prior / non-concurrent works as you have done with [5]. Finally, [2,4] have explored the **exact same PINN-based objective** for learning the vector field (and much more thoroughly so from an empirical standpoint), so I'm not sure from a methods standpoint what contribution on the methods side this work brings.

 It is a shame; the paper is very well written, and it is a good idea. However, it has indeed been already explored. I am happy to focus more on the theoretical contributions in the discussion period, and given a better understanding of their novelty I will be happy to increase my score; in particular, I suspect the assumptions in proposition 2 of [1] do seem (Assumption D1) like they might be more restrictive and their results seem to have a less broader scope than yours (i.e. you show the curves are a.c. in the Wasserstein 2 space, which feels like a stronger and more insightful result).

Another point that seems novel is the gradient flow interpolation from what I can see in works [1,2,4] they also explored learned interpolations too and in particular the $t(1-t) * \mathrm{NN}(x,t)$ parameterizations can be seen in [2,4,5], that said the gradient flow interpolation seems novel, If a more comprehensive comparison of this to prior works / motivation as to why this is an improvement over [1,2,4]  is provided I would also happily increase my score.


[1] Nusken, N., Vargas, F., Padhy, S. and Blessing, D., 2024, May. Transport meets variational inference: Controlled monte carlo diffusions. In The Twelfth International Conference on Learning Representations: ICLR 2024.

[2] Sun, J., Berner, J., Richter, L., Zeinhofer, M., Müller, J., Azizzadenesheli, K. and Anandkumar, A., 2024. Dynamical measure transport and neural pde solvers for sampling. arXiv preprint arXiv:2407.07873.

[3] Albergo, M.S. and Vanden-Eijnden, E., 2024. NETS: A Non-Equilibrium Transport Sampler. arXiv preprint arXiv:2410.02711.

[4] Sun, J., Berner, J., Azizzadenesheli, K. and Anandkumar, A., Physics-informed neural networks for sampling. In ICLR 2024 Workshop on AI4DifferentialEquations In Science.

[5] Máté, B. and Fleuret, F., 2023. Learning interpolations between boltzmann densities. arXiv preprint arXiv:2301.07388.

**Questions:**

Am I correct in understanding that the linear and learned methods in your experiments are no different to the approach proposed in [5] ? thus from a methods standpoint the gradient flow interpolation is the novel algorithmic contribution ?

---

> ### Author Response · Authors · 2024-11-18
>
> Many thanks  for your careful and detailed review, in particular for pointing to additional references! We will incorporate them in the revised version. Below we address your concerns.
>
> In our work we fundamentally consider two different situations. First, we assume that we are given a prescribed path $\rho_t$ **with tractable** $\log \rho_t$ (up to a constant). Then we consider learning a vector field $v_t$ which governs the evolution of $\rho_t$ via the continuity equation. This is also the focus of the works [1,3] and leads naturally to the theoretical question for which curves of densities $\rho_t$  a vector field $v_t$ with suitable properties exists. More precisely this leads to the question of existence of a solution to the Poisson equation
> $\nabla\cdot(\rho_t\nabla \phi_t) =\rho_t(\alpha(t,\cdot)-\mathbb E_{\rho_t}[\alpha])$. We interpret the Poisson equation as combining the Wasserstein geometry on the left and the Fisher Rao geometry on the right. There are three very important differences between the treatment of [1] and our theory.
>
> - They assume that $\rho_t\in C^\infty([0,1],\mathbb R^n)$ and $\alpha\in C^\infty([0,1]\times \mathbb R^n)$ and a Poincare inequality for  **each time** $\rho_t$. We need  **much weaker** conditions: we assume that $t\mapsto\rho_t$ is weakly continuous,  $\rho=\int \rho_t dt$ fulfills a (partial in $x$) Poincare inequality and $\alpha\in L^2(\rho)$.  In particular, we allow single time points $t$, where the Poincare inequality might fail for $\rho_t$.
> - Our solution $\phi$ is an element of $H^1(\boldsymbol{\rho})\subset L^2([0,1]\times \mathbb{R}^d; \boldsymbol{\rho})$ and thus automatically measurable  on the **whole** time domain $t \in [0,1]$. This makes it possible to show that $\nabla \phi_t(x)$ is a Borel vector field  with regularity in time which implies absolutely continuity  of the flow $\rho_t$ in the Wasserstein geometry. In contrast, [1] only considers a solution  only for every **single** $t\in[0,1]$ without any regularity in $t$. This is a result already shown in [laug] Theorem 2.2 (cited by us) and motivated our investigation into a framework across the time $t$.
> - Furthermore, we give precise mild assumption under which the partial Poincare inequality is satisfied. In particular, we cover the case of the linear interpolation. As a result we obtain that these curves are absolutely continuous (AC) in the Wasserstein geometry.
>
> Hence our work gives a rigorous theoretical foundation which can be applied to the setting in [1]. For linear interpolation of energies which is of interest for many authors, we do not only show the existence of a suitable vector field, but also provide a **nontrivial analytical example**, where the corresponding vector field becomes exponentially irregular, see line 268 and Appendix A4.
>
>
> Second, motivated by our above findings, we consider approaches to learn more regular curves **without  tractable** $\log \rho_t$. The method proposed by [5] adds a learned regularizer $t(1-t) \psi_t$ to the linear interpolation. This is exactly what we call learned interpolation
> and used for comparisons (see your question).
> We instead consider the case of learning a regular AC curve directly by considering a Wasserstein gradient flow in the Fisher Rao geometry. When finishing this paper, we became aware of [2], we thank the reviewer for pointing out [4] which we were not aware of. In [2,4] the authors make similar considerations but coming from an SDE based stochastic optimal control perspective **without considering the relation to Fisher Rao and Wasserstein geometry**. While we simulate the corresponding ODE instead of the SDE and use a different sampling strategy for the PDE collocation points, eventually,  up to time reparametrization, we end up with the same loss (see your question).
>
>
> The advantage of using the gradient flow interpolation is that we know the resulting  curve whose favorable properties were extensively studied  in the literature. This opens the door for further analysis. In contrast, when learning the curve with learned regularizer [5], there is no knowledge of the resulting curve.
>
>
> [laug]: R. S. Laugesen, P. G. Mehta, S. P. Meyn, and M. Raginsky. Poisson’s equation in nonlinear filtering.
> arXiv preprint arXiv:1412.5845, 2014.

---

> > ### Comment · Reviewer_m4s7 · 2024-11-19
> > **Thanks for clarifying score raised**
> >
> > Thank you for your clarification, with some of the details you have provided I can see how the theoretical results are stronger than those in [1] and I have raised my score to reflect this, however please do include a detailed comparison/discussion (as you have in the rebuttal) in the final revised version.
> >
> > I think the manuscript would benefit from detailing more explicitly what contribution 3. in the introduction achieves in practice , in other works the non-smooth transport is often referred to a mode-switching and I completely missed you were addressing this on my first pass of the paper as its described very quickly/in passing.
> >
> > Note, there are some computational fallbacks arising from this parameterization, for example one now has to compute a Laplacian of a network wrt to its inputs which is very expensive in high dimensions although you could potentially argue its not much more expensive than the already required divergences by any PINN loss on continuity, still these terms are quite difficult for high dim sampling tasks, there's not much evaluation, discussion of this.
> >
> > > Second, motivated by our above findings, we consider approaches to learn more regular curves without tractable
> >
> > This is not entirely novel again if you inspect [2,4] you will see they have several PINN-based objectives for which they require learning both the score and the velocity field as the density is unknown/non-tractable , see their $\mathcal{L}_{\mathrm{log-CE}}$ loss for example.
> >
> > In [2,4] their SDE-Score objective $\mathcal{L}_{\mathrm{score}}$ is learning a VP-SDE based spatial interpolation) which is probably worth mentioning in much more detail in the appendix; you already discuss this in your response to my questions the main differences being SDE vs ODE and the perspectives from which the ideas are developed but ultimately the same velocity field is being learned and it also corresponds to the same flows/conitinuity eq as in methods such as [6] and [7] (note [6] mentioned the ODE/vector field param too in the context of sampling, once you have access to the score the SDE-ODE conversion in this case is straightforward) which should be discussed in more detail, whats the advantage of a pinn/ODE approach over the SDE approaches already explored to learn this flow and ultimately given the closeness in the approaches an empirical comparison / ablation between them feels needed.
> >
> > Finally, I still think this paper is missing evaluation and comparison; whilst it is good to compare to two other interpolations and showcase an improvement from an empirical side it feels only half-way there, showcasing behaviour in dimensions higher than 2 and 8 which even for sampling are very small feels very limited.
> >
> > Finally showcasing how these methods compare to some of the other recent SDE/ODE based samplers, or other ways to learn continuity such as [1] would be more helpful to see how the method places.
> >
> > **Q:**  You discuss the relationship between Equation 18 and Wasserstein gradient flows, and you also discuss the advantage of spatial interpolations. But you never clearly state that the minimiser of  $\mathcal{E}(\theta,x,t)$ under the proposed parameterization corresponds to the VP-SDE spatial interpolation ?  if I look at the objective from the SDE perspective I can see eq 18 just becomes a log-FPK equation for an OU process given a Gaussian starting distribution, as you mention it yields the same objective as in [2,5], so its easy to see the correspondence, but it would be nice if you could state this more clearly/formally, you kinda discuss it and maybe allude to it in Remark 3 ? but I cant really see a clear statement of this.
> >
> > Whilst I appreciate the technical (theoretical) contributions and perspectives, I still do not remain convinced that the new proposed parametrisation /objective is novel (as you have pointed out, in the end, it coincides with  $\mathcal{L}_{\mathrm{score}}$ from [2,4], this and the minimal experiments are the reasons for the borderline score.
> >
> > [6] Vargas, F., Grathwohl, W. and Doucet, A., 2023. Denoising diffusion samplers. arXiv preprint arXiv:2302.13834.
> >
> > [7] Berner, J., Richter, L. and Ullrich, K., 2022. An optimal control perspective on diffusion-based generative modeling. arXiv preprint arXiv:2211.01364.

---

> > > ### Author Response · Authors · 2024-11-21
> > >
> > > Thank you again for your very in-depth response!
> > >
> > > - Following your suggestion, we added a discussion of our theoretical results at the end of Appendix A, comparing our theory with the one developed in [1, laug].
> > >
> > > - Regarding the mode switching phenomena we included Example 5 which showcases a configuration where the linear interpolation fails due to mode switching while the gradient flow interpolation correctly approximates both target modes. Thank you also for pointing out the term 'mode switching', we included it in the respective section.
> > >
> > > - It is true that the computation of the divergence/ Laplacians can become prohibitive when scaling to very high dimensions. We did not discuss this in the first version, but added a short remark in our limitations section now.
> > >
> > > - We have added further details in the related work section concerning [6,7]. Furthermore, we added clarification on the relation of the gradient flow interpolation and the OU process. In particular we discussed the relation to the loss $\mathcal{L}_{\text{score}}$ from [2,4]. Regarding your question we also clarified that the minimizer of our loss corresponds to the log density (up to a constant) of the solution to a VP-SDE.
> > >
> > > Finally we would like to thank you for your constructive feedback, our manuscript benefited greatly from your input.

---

> > > > ### Comment · Reviewer_m4s7 · 2024-11-22
> > > > **Thanks for the revised version**
> > > >
> > > > The revised version reads much more clearly, and on a second pass, I think it's much easier to appreciate the conceptual contributions of this paper, thus I have updated my score. I think more so than being a paper that proposes a new method, this paper sheds a much more technical light on the issue of mode switching, which in most prior works I am familiar with have mostly provided an empirical analysis (some references for the term mode/label switching [7,8, 9, 10, 11]).
> > > >
> > > > Some final points that are worth discussing in the appendix final version:
> > > >
> > > > 1. An alternative to a VP-SDE based flows  (such as the one studied here and approaches such as DDS and PIS) is the time reversal of a pinned Brownian motion [12,13,14]  (also a spatial flow) here a flow that interpolates between a point mass $\delta_0$ and the target distribution, in  Corollary 2 of [15] shows how the score/drift becomes singular for a very simple gaussian target similar to the more general example you discuss for the linear interpolation.  It would be interesting to see how this flow compares, as it is possibly the most used learning flow from these variational/learned approaches to sampling. Note this flow has no ODE counterpart, only FPK no continuity.
> > > > 2. Whilst VP-SDE / spatial interpolation-styled flows address the mode-switching problem, they do introduce an ergodicity error that is unlike the linear or the learned flows, which reach the target distribution at time $t=1$ the OU process never converges to a Gaussian and thus $\mathrm{Law}X_0 \approx \mathcal{N}(0,I)$ and thus it incurs an error proportional to $e^{-1}$ or $e^{-\int_0^1 \beta_s \mathrm{d}s}$ due to the fast mixing of the OU process, in contrast with the fixed (linear or learned) interpolations which have no mixing error, so its worth maybe acknowledging this as a potential fallback in contrast to other "exact" / "finite"-time methods.
> > > >
> > > > Minor typographic comment: The bib-ref for [1] seems to have made the first author second despite the ordering on the ICLR version of the paper, maybe worth ammending.
> > > >
> > > >
> > > > [7] Dawn Woodard, Scott Schmidler, and Mark Huber. Sufficient Conditions for Torpid Mixing of
> > > > Parallel and Simulated Tempering. Electronic Journal of Probability, 14(none):780 – 804, 2009.
> > > > doi: 10.1214/EJP.v14-638. URL https://doi.org/10.1214/EJP.v14-638.
> > > >
> > > > [8] Syed, S., Bouchard-Côté, A., Deligiannidis, G. and Doucet, A., 2022. Non-reversible parallel tempering: a scalable highly parallel MCMC scheme. Journal of the Royal Statistical Society Series B: Statistical Methodology, 84(2), pp.321-350.
> > > >
> > > > [9] Phillips, A., Dau, H.D., Hutchinson, M.J., De Bortoli, V., Deligiannidis, G. and Doucet, A., 2024. Particle Denoising Diffusion Sampler. arXiv preprint arXiv:2402.06320.
> > > >
> > > > [10] Chehab, O. and Korba, A., 2024. A Practical Diffusion Path for Sampling. arXiv preprint arXiv:2406.14040.
> > > >
> > > > [11] Noble, M., Grenioux, L., Gabrié, M. and Durmus, A.O., 2024. Learned Reference-based Diffusion Sampling for multi-modal distributions. arXiv preprint arXiv:2410.19449.
> > > >
> > > > [12] Zhang, Q. and Chen, Y., 2021. Path integral sampler: a stochastic control approach for sampling. arXiv preprint arXiv:2111.15141.
> > > >
> > > > [13] Vargas, F., Ovsianas, A., Fernandes, D., Girolami, M., Lawrence, N.D. and Nüsken, N., 2023. Bayesian learning via neural Schrödinger–Föllmer flows. Statistics and Computing, 33(1), p.3.
> > > >
> > > > [14] Reu, T., Vargas, F., Kerekes, A. and Bronstein, M.M., To smooth a cloud or to pin it down: Expressiveness guarantees and insights on score matching in denoising diffusion models. In The 40th Conference on Uncertainty in Artificial Intelligence.
> > > >
> > > > [15] Vargas, F., Grathwohl, W. and Doucet, A., 2023. Denoising diffusion samplers. arXiv preprint arXiv:2302.13834.

---

> > > > > ### Author Response · Authors · 2024-11-25
> > > > >
> > > > > Once again we thank you for your considerable input. We have included the references concerning mode switching. Furthermore we have added a small remark on the fact that the gradient flow interpolation incurs a mixing error. With respect to the flows stemming from a time reversal of a pinned Brownian motion we updated our related work section. We also find the relation between these flows and the phenomenon described in [15] very interesting and we will explore this further. We have also corrected our typographic error in [1], thanks for pointing it out!

---

### Meta-Review · Area_Chair_PLMm · 2024-12-20

**Metareview:**

The authors studied Fisher--Rao flows from both a theoretical and an algorithmic point of view. In particular, on a conceptual level, the paper has made clear contributions towards understanding when a flow satisfies the continuity equation, equivalently absolutely continuous in Wasserstein geometry. Furthermore, the authors provided a clear example towards understanding the problems with linear interpolation leading to an exploding norm, which has practical impacts in algorithm design.

Despite the somewhat borderline scores, a very productive discussion period clarified the contributions of this work. More specifically, the goal of this paper was not to propose a new algorithm yielding directly performance improvements, but rather understanding a common issue for a large class of flow related algorithms. For these reasons, I would recommend accept for this paper.

**Additional Comments On Reviewer Discussion:**

Reviewer m4s7 had a particularly productive discussion period with the authors. Several issues of novelty were clarified, as the authors did not claim novelty of the algorithms, but rather the analysis of absolute continuity and its consequences. For me, this was very help to understanding the context of this paper within the existing literature, and see the potential impact for the future.

---

### Decision · Program_Chairs · 2025-01-22

Accept (Poster)